# Accelerating hybrid XOR–CNF Boolean satisfiability problems natively with in-memory computing

Haesol Im [1,7], Fabian Böhm [2,7], Giacomo Pedretti [3], Noriyuki Kushida[1], Moslem Noori [1], Elisabetta Valiante [1], Xiangyi Zhang[1], Chan-Woo Yang [1], Tinish Bhattacharya [4], Xia Sheng[3], Jim Ignowski [3], Arne Heittmann[5], John Paul Strachan [5,6], Masoud Mohseni[3], Raymond Beausoleil[3], Thomas Van Vaerenbergh [2] & Ignacio Rozada [1] ✉

The Boolean satisfiability (SAT) problem is a computationally challenging decision problem central to many industrial applications. For SAT problems in cryptanalysis, circuit design, and telecommunication, solutions can often be found more efficiently by representing them with a combination of exclusive OR (XOR) and conjunctive normal form (CNF) clauses. We propose a hardware accelerator architecture that natively embeds and solves such hybrid XOR–CNF problems using in-memory computing hardware. To achieve this, we introduce an algorithm and demonstrate, both experimentally and through simulations, how it can be efficiently implemented with memristor crossbar arrays. Compared to the conventional approaches that translate XOR–CNF problems to pure CNF problems, our simulations show that the accelerator improves computation speed, energy efficiency, and chip area utilization of in-memory accelerators by ∼10× for a set of hard cryptographic benchmarking problems. Moreover, the accelerator achieves a ∼10× speedup and a ∼1000× gain in energy efficiency over state-of-the-art SAT solvers running on CPUs.

The Boolean satisfiability (SAT) problem is a fundamental decision problem that was the first problem to be proven NP-complete[1,2]. Solving a SAT problem involves determining whether there is an assignment of Boolean variables satisfying a given propositional logic formula. Many problems in engineering and computer science reduce to SAT problems with a polynomial-time overhead, which then can be tackled with SAT solvers employing local search heuristics or exhaustive search. SAT solvers are thus widely employed in many industry-relevant applications, such as scheduling, planning, cryptanalysis, and integrated circuit design[3,4], as well as being used as the engine for more-general constrained optimization solvers[5]. Yet, due to the computational complexity of SAT problems, the cost of finding solutions

could, in the worst case, scale exponentially with the number of variables.

Due to the ubiquity of SAT problems in industrial optimization applications, there is an ongoing effort to improve algorithms for SAT solvers, as well as to develop dedicated hardware accelerators[6–14] that can find solutions faster and more energy efficiently. A promising line of research has been the study of SAT solvers in hybrid problem formulations[15–17]. SAT problems are typically formulated in conjunctive normal form (CNF), where a set of clauses containing Boolean variables are connected by logical OR operations. However, many applications naturally involve clauses linked by exclusive-OR (XOR) operations, such as channel decoding in wireless receivers[18], model counting[15],

[1]1QB Information Technologies (1QBit), Vancouver, BC, Canada. [2]HPE Labs, Hewlett Packard Enterprise, Brussels, Belgium. [3]HPE Labs, Hewlett Packard Enterprise, Milpitas, CA, USA. [4]University of California, Santa Barbara, CA, USA. [5]Peter Grünberg Institute (PGI-14), Forschungszentrum Jülich GmbH, Jülich, Germany. [6]RWTH Aachen University, Aachen, Germany. [7]These authors contributed equally: Haesol Im, Fabian Böhm. ✉e-mail: ignacio.rozada@1qbit.com

circuit fault testing[3], and cryptographic decoding attacks[19]. These problems can be formulated natively as hybrid XOR−CNF SAT problems containing both CNF and XOR clauses. Although XOR clauses can be reduced to CNF clauses using Tseitin transformations[20], doing so introduces a significant performance overhead as it increases the number of variables and clauses in the problem. Hybrid XOR−CNF SAT solvers that support both CNF and XOR clauses have therefore been found to considerably outperform pure CNF SAT solvers[17,21].

While hybrid XOR−CNF solvers have predominantly been implemented as software solutions running on digital computers[16,22], there is potential in harnessing the benefits of native XOR−CNF problem formulations using in-memory hardware accelerators. In-memory computing (IMC), leveraging analog crossbar arrays for low latency and parallel linear algebra computations, is a promising technology for building hardware accelerators[23]. IMC accelerators have already demonstrated their ability to enhance both speed and energy efficiency for SAT solvers in the case of pure CNF SAT problems, outperforming conventional CPUs[8,9,24]. Combining the advantages of a hybrid XOR−CNF formulation with IMC hardware could offer considerable advantages in tackling computationally challenging SAT problems with inherent XOR clauses. However, compared to pure CNF problems, evaluating XOR clauses requires more complex and energy-intensive circuits that can potentially offset the efficiency and latency advantages of IMC hardware. Moreover, XOR clauses can contain many literals, whereas SAT hardware accelerators can often support only a few literals per clause. For IMC hardware, a large number of literals can also make it more challenging to retain low error rates during computation, as the corresponding analog signals exhibit an increased dynamic range.

Therefore, in this work, we set to address the open question of whether IMC is suitable for accelerating the solving of hybrid XOR−CNF problems efficiently. We present an IMC accelerator architecture that can be used to natively implement and solve hybrid XOR−CNF problems. As part of this architecture, we propose Walk-SAT-XNF, an XOR-native implementation of the WalkSAT stochastic local search (SLS) heuristic, where all variables within unsatisfied clauses are candidates for being flipped. We propose an efficient method for XOR−CNF clause evaluation and gradient computation using analog crossbar arrays. To demonstrate feasibility on hardware, we experimentally implement WalkSAT-XNF on crossbar arrays based on TaO$_x$ memristors for a small-scale minimal disagreement parity (MDP) problem. Additionally, we simulate a memristor-based accelerator architecture in a 28 nm complementary metal-oxide-

semiconductor (CMOS) process and evaluate the computation speed and energy consumption on benchmarking problems from cryptographic applications including the McEliece−Niederreiter cryptosystem[25,26] and the Advanced Encryption Standard (AES)[27,28]. Compared to solving problems in their CNF representation with an IMC accelerator, our approach achieves an order-of-magnitude improvement in computation speed and energy consumption, within a 10 × smaller chip area, by employing hybrid XOR−CNF representations. Furthermore, compared to state-of-the-art SAT solvers running on CPUs, our accelerator solves benchmarking problems with up to 300 variables and 1016 clauses ~10 × faster while consuming ~1000 × less energy. Our results highlight the potential of IMC accelerators for efficiently implementing hybrid XOR−CNF SAT solvers, enabling native problem representations for solving a variety of complex industry-relevant problems.

## Results
### Mapping and benchmarking advantages of hybrid XOR−CNF SAT problems over CNF

A SAT problem for a set of Boolean variables $x_i \in \{0, 1\}$ and clauses $C_i$ is given by the conjunction ($\vee$)

$$\mathcal{F}(x_1, \ldots, x_n) = C_1 \wedge C_2 \wedge \cdots \wedge C_i. \quad (1)$$

The problem is said to be satisfiable if an assignment of the Boolean variables exists where all clauses $C_j$ are true. In a CNF representation, each $C_j$ is a clause formed from a disjunction ($\wedge$) of literals $l_k$ as $C_{\mathrm{CNF},j} = l_k \wedge \cdots \wedge l_m$, where the literals $l_k$ are either propositions ($x_k$) or their negations ($\overline{x_k}$) of the Boolean variables. XORSAT problems, on the other hand, are SAT problems where clauses are formed using XOR operations ($\oplus$) between literals:

$$C_{\mathrm{XOR},j} = l_k \oplus \cdots \oplus l_m.$$

Problems formulated in XOR-and-OR normal form (XNF) are then hybrid XOR−CNF SAT problems, where the propositional logic formula (1) contains both CNF and XOR clauses. Figure 1a illustrates an XNF instance with three CNF and two XOR clauses. Here, the variable assignment $x_1 = 1$, $x_2 = 0$, $x_3 = 0$, $x_4 = 1$ guarantees satisfiability. In general, an XOR clause with $k$ literals $x_1, \ldots, x_k$ can be equivalently represented using $2^{k-1}$ CNF clauses, each containing $k$ literals. These clauses represent all possible combinations of an even number of

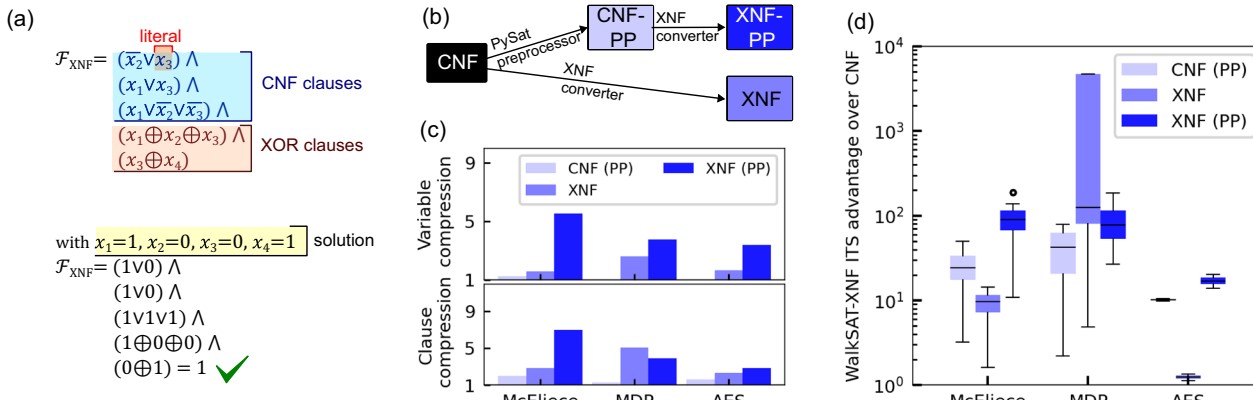

**Fig. 1 | Mapping advantages of hybrid XOR−CNF problems over pure CNF problems. a** XNF SAT instance containing CNF and XOR clauses and a solution that certifies its satisfiability. **b** Strategies for converting CNF instances to XNF instances. **c** Average variable and clause compression ratio obtained by the preprocessing strategies on three classes of XNF problems. Error bars show the standard deviation. **d** Advantages in iterations-to-solution for the WalkSAT-XNF heuristic when comparing different problem representations to the CNF formulation. The box-and-whisker plots shows the median and interquartile range.

**Table 1 | WalkSAT-XNF heuristic**

| | |
|---|---|
| 1: | **function WalkSAT-XNF**noise_level, clauses, max_iter |
| 2: | configuration ← assign binary values |
| 3: | iter ← 0 |
| 4: | **while** iter ≤ max_iter **do** |
| 5: | $\mathcal{U}$ ← { variable : variable *in* unsatisfied clauses} |
| 6: | **for** variable ∈ $\mathcal{U}$**do** |
| 7: | gain$_{variable}$ ← COMPUTE_GAIN_VALUE(variable, configuration, clauses) |
| 8: | noisy_gain$_{variable}$ ← gain$_{variable}$ + noise_level · $e$, $e \sim \mathcal{N}(0,1)$ |
| 9: | **end for** |
| 10: | variable_to_flip ← arg max {noisy_gain$_{variable}$ : variable ∈ $\mathcal{U}$} |
| 11: | configuration[variable_to_flip] ← **flip** configuration[variable_to_flip] |
| 12: | **if** all clauses evaluated at configuration are satisfied **then** |
| 13: | **return** TRUE ▷ The instance is satisfiable |
| 14: | **end if** |
| 15: | iter ← iter + 1 |
| 16: | **end while** |
| 17: | **return** FALSE ▷ Solution is not found |
| 18: | **end function** |
| 19: | **function** Compute_Gain_Valuevariable, configuration, clauses |
| 20: | $\mathcal{C}$ ← clauses |
| 21: | break_count ← 0 |
| 22: | make_count ← 0 |
| 23: | **for** C ∈ { clause : clause in $\mathcal{C}$ connected to variable}**do** |
| 24: | N ← number of true literals in C evaluated at configuration |
| 25: | **if** C is CNF clause **then** |
| 26: | **if** N = 0 **then** |
| 27: | make_count ← make_count + 1 |
| 28: | **end if** |
| 29: | **if** N = 1 **then** |
| 30: | break_count ← break_count + 1 |
| 31: | **end if** |
| 32: | **else if** C is XOR clause **then** |
| 33: | **if** N is even **then** ▷ Currently violated |
| 34: | make_count ← make_count + 1 |
| 35: | **else if** N is odd **then** ▷ Currently satisfiable |
| 36: | break_count ← break_count + 1 |
| 37: | **else if** |
| 38: | **else if** |
| 39: | **end for** |
| 40: | **return** make_count − break_count |
| 41: | **end function** |

negated variables

$$C_{\text{XOR},j} = \bigwedge_{\text{even number of}} \pm x_1 \vee \cdots \vee \pm x_k, \qquad (2)$$

where ± denotes the possible permutations for propositions (+) of literals or their negations (−). For instance, the first XOR clause in Fig. 1a has the equivalent CNF representation $(\overline{x_1} \vee \overline{x_2} \vee x_3) \wedge (\overline{x_1} \vee x_2 \vee \overline{x_3}) \wedge (x_1 \vee \overline{x_2} \vee \overline{x_3}) \wedge (x_1 \vee x_2 \vee x_3)$. Translating XOR clauses into CNF clauses incurs an exponential increase in the number of additional clauses, hence making clause evaluation computationally more expensive.

In practice, this exponential overhead can partly be mitigated by employing the Tseitin transformation[20], yet this method provides a clear trade-off between the reduction of overall clauses and the

number of additional variables that need to be considered[16]. Conversely, translating a SAT problem in CNF representation into an XORSAT problem is generally impossible, though many key SAT applications, such as integer factorization, circuit fault testing[4], and cryptographic decoding attacks[19], originate from XOR-based logic. In these cases, XOR clauses can be reconstructed from the CNF clauses by reversing the transformation in Eq. (2), typically reducing both clause and variable counts.

We demonstrate the differences between CNF and XNF formulations in Fig. 1 for SAT problems from cryptographic attacks on the McEliece–Niederreiter and AES cryptosystems, as well as instances generated from the minimal disagreement parity (MDP) problem (details of the instances are provided in the Methods section). All instances inherit native XOR clauses but are initially provided with CNF clauses only. We explore two methods of generating hybrid XOR–CNF instances from these original problems. First, we convert directly the CNF instances to the XNF representation employing the cnf2xnf tool within the xnfSAT solver[16]. The final representation of this process is denoted by XNF in Fig. 1b. After this conversion, the resulting problems contain 2–43% XOR clauses. Additionally, we employ a SAT preprocessing (PP) tool[29] to the CNF instances (generating new instance denoted by CNF-PP in Fig. 1b) before applying the conversion tool to generate XNF instances. The final representation of this process is denoted by XNF-PP in Fig. 1b. Such preprocessing techniques are widely used to compress CNF problem size and to enhance solver performance. Details of the preprocessing procedure and the per-instance preprocessing runtime are reported in Section "Methods". Figure 1c shows the compression ratio for the number of variables in relation to the original CNF representation. Direct XNF conversion reduces the number of variables by (2.0 ± 0.5)× on average. When applying preprocessing, the average number of variables initially remains almost unchanged ((1.1 ± 0.1)×) but is considerably reduced once the problem has been converted to an XNF representation. The preprocessing followed by XNF conversion achieves a compression ratio of (4.6 ± 1.0)×, on average. We also analyze the compression ratio for the number of clauses in relation to the CNF representation. With direct XNF conversion, we find that the number of clauses is reduced by (3.7 ± 1.2)×, on average. When applying preprocessing to the CNF representation, we again observe a small initial reduction in the number of clauses by (2.0 ± 0.9)×, while conversion of the pre-processed instances to an XNF representation reduces the number of clauses by (5.4 ± 1.8)×, on average, compared to the CNF representation.

These results show the advantages of mapping problems to an XNF representation, with the greatest benefits often observed when combining preprocessing with XNF conversion. Compared to using a pure CNF representation, the resulting reduction in the problem size can enhance SAT solver performance and significantly lowers compute resource requirements[17,21]. Moreover, for SAT hardware accelerators, the comparatively smaller XNF instances enable reduced chip sizes and energy consumption. Therefore, these results serve as a strong motivation to develop hardware accelerators capable of supporting both CNF and XOR clauses simultaneously.

## WalkSAT-XNF: an XNF-native SAT heuristic compatible with in-memory computing hardware

To leverage the described mapping advantages, we propose a heuristic called WalkSAT-XNF, designed to solve XNF problems in their native form. We then show how this algorithm can be realized efficiently in an accelerator using IMC. WalkSAT-XNF employs a local search heuristic and is inspired by prior work on IMC accelerators for CNF SAT problems[8]. Similar to the widely used WalkSAT solvers[30,31], WalkSAT-XNF computes gradients based on 'make' and 'break' values. The make value counts the number of violated clauses that become satisfied, while the break value counts the number of satisfied clauses that

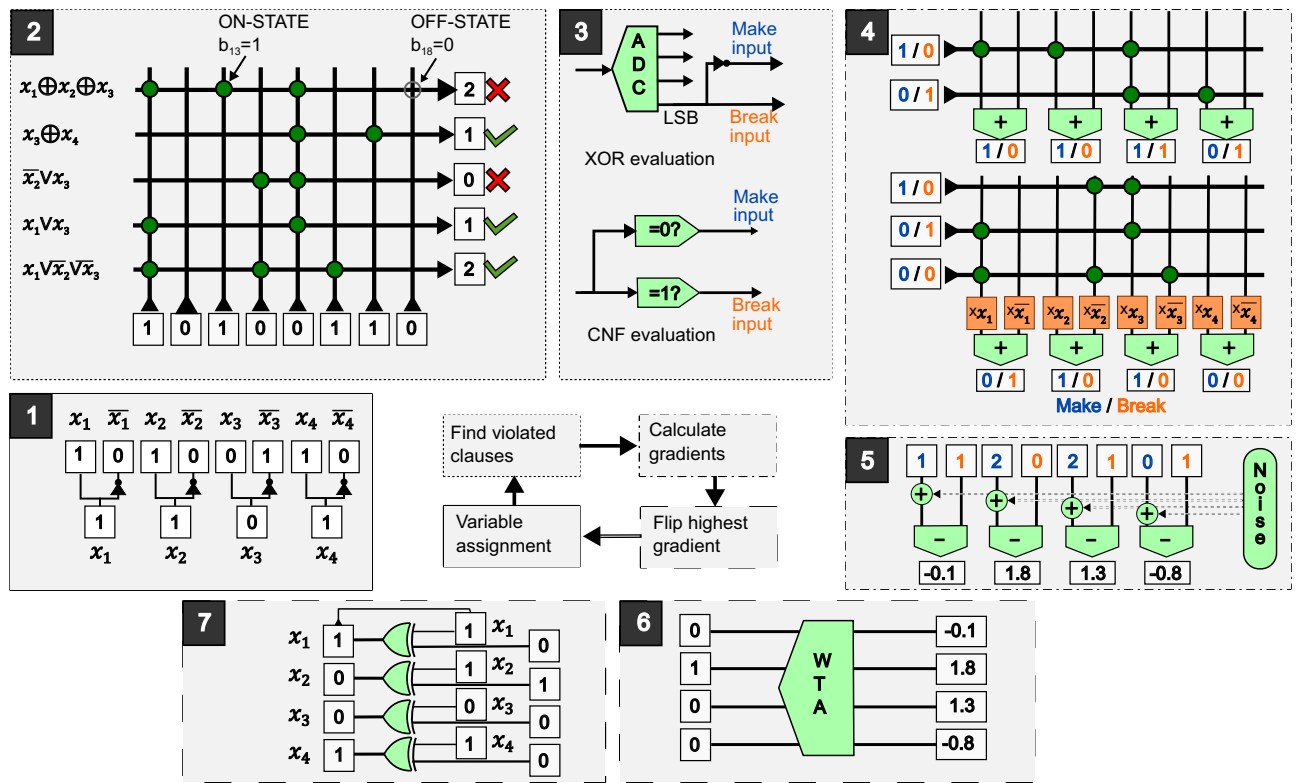

**Fig. 2 | Hardware architecture for an in-memory XOR−CNF solver accelerator.** Hardware architecture for implementing WalkSAT-XNF with IMC. An iteration of WalkSAT-XNF is performed sequentially by a register (1), a clause lookup crossbar array (2), clause evaluation circuits (3), a make and break computation crossbar array (4), a gradient computation (5), a winner-takes-all circuit (6), and a variable flip (7). The function of these elements is shown for the example SAT problem in Fig. 1a and an initial variable assignment $x_1 = 1$, $x_2 = 1$, $x_3 = 0$, and $x_4 = 1$.

become violated when flipping a variable. WalkSAT-XNF then flips a variable found in violated clauses that maximizes the value obtained by subtracting the break value from the make value. In contrast to the standard WalkSAT heuristic, WalkSAT-XNF performs a full-neighborhood evaluation, where gradients for all variables present in unsatisfied clauses are considered, as opposed to evaluating only the variables in a randomly chosen violated clause.

Table 1 shows the pseudocode of the WalkSAT-XNF heuristic. The algorithm starts with an initial variable configuration and iteratively searches the space until it finds a solution or reaches the iteration limit. Each iteration computes gradients based on make and break values for all variables by evaluating the clauses in which they appear. A CNF clause is satisfied if at least one literal is true. Hence, the make value is the number of violated clauses containing the variable, as flipping it would satisfy them. The break value, on the other hand, corresponds to the number of satisfied clauses, where the variable is the only true literal, as flipping it would break clause satisfaction. For an XOR clause to be satisfied, an odd number of true literals is required. Thus, the make value corresponds to the number of violated clauses containing the variable, as flipping it would satisfy them. Similarly, break values are equal to the number of satisfied clauses containing the variable. The break value subtracted from the make value yields the gain value, or gradient. After computing the full gradient, Gaussian noise with a standard deviation $\sigma$ is added to help escape local minima or avoid cycles. The variable with the highest noise-adjusted gain value is then flipped, and the process repeats.

Figure 1d shows the algorithmic efficiency of WalkSAT-XNF when solving the McEliece, MDP, and AES benchmarking instances using CNF-PP, XNF, and XNF-PP compared to the CNF formulation. We quantify the performance with the iterations-to-solution (ITS$_{99}$)

metric[32], defined as

$$\text{ITS}_{99}(\text{iter}) := \frac{\text{iter} \cdot \log 0.01}{\log(1 - \theta(\text{iter}))}, \qquad (3)$$

where $\theta(\text{iter})$ is the success probability of solving the problem as a function of iterations. The ITS$_{99}$ metric estimates the iterations required to observe at least one successful trial with a probability of 99%. Since WalkSAT-XNF stops once a solution is found, an optimized ITS$_{99\text{opt}}$ metric can be obtained by evaluating ITS$_{99}$ at solution-finding trial lengths within reasonable error bounds. Compared to the CNF formulation, WalkSAT-XNF solves problems using fewer iterations, achieving a median improvement of ~23× (CNF-PP), ~10× (XNF), and ~68× (XNF-PP). The greatest performance gains are observed for preprocessed instances.

In what follows, we thus solely focus on the preprocessed instances for CNF and XNF problems, referring to them simply as CNF and XNF for brevity. Complete benchmarking results for all problem representations are available in Supplementary Note 1.

## An in-memory computing accelerator architecture for WalkSAT-XNF

To realize WalkSAT-XNF with IMC hardware, we propose the accelerator architecture depicted in Fig. 2, which shows the steps performed in each iteration of the heuristic (i.e., clause evaluation, make and break value computations, and a variable update) using seven distinct hardware blocks.

The Boolean variable configuration is initially stored in a register ((1) in Fig. 2). The variables and their respective conjugates are then provided as an input signal to a crossbar array to evaluate violation of

the individual CNF and XOR clauses (2). For problems with $N$ variables and $C$ clauses, the crossbar has $2N$ columns and $C$ rows. The input to the crossbar is applied as binary voltage signals at the columns. Each variable $x_j$ and its negation $\overline{x_j}$ are mapped to the column pairs $\{2j, 2j + 1\}$, while clauses correspond to the rows of the crossbar. Each literal is represented by a binary-valued crossbar connection $b_{ij} \in \{0, 1\}$ that allows current to flow from a column to a row. Here, positive literals $x_j$ connect rows to columns with even indices $2j$, while negative literals $\overline{x_j}$ connect to columns with odd indices $2j + 1$. These connections are facilitated by memory devices at each crossbar that can be switched between an ON and an OFF state, such as resistive random-access memory (RRAM)[8], static random-access memory (SRAM), or embedded Flash memory cells[33]. This crossbar array functions as a $C$-by-$2N$ matrix, with entries of 1 where literals appear and 0 elsewhere. The output current at each row is then equivalent to a matrix–vector multiplication between the input signal and the array. Using the matrix encoding of the clauses described above, the output signals of the crossbar rows are proportional to the number of true literals in the clauses for the current assignment of variables.

Depending on the clause type, the output signals from the crossbar array are evaluated by the circuits (3) of Fig. 2. These circuits indicate whether a clause is violated and provide the input signals for the subsequent make and break value computations. For XOR clauses, a low-resolution analog-to-digital converter (ADC) with $\log_2(k)$ bits, where $k$ is the maximum number of literals, performs a parity check using the least-significant bit (LSB). The LSB is provided as input for the break value computation, as it indicates whether the clause is currently satisfied and can be broken by flipping one of its member variables. Conversely, an inversion of the LSB is given as input for the make value computation. For CNF clause evaluations, two comparators[8] determine if the number of true literals is 0 (for the make value) or 1 (for the break value). The outputs of these comparators are used as input for make and break computations.

The make and break values are computed via a crossbar array (4) that is the transpose of (2). After applying the input signals to the rows, the output signals from related pairs of columns are added to derive the make and break values for each variable. To calculate the break values for CNF clauses, the column outputs are additionally multiplied with the variable configuration using pass transistors to identify true literals. Adding the make and break values from XOR and CNF clauses provides the input signals for the subsequent gradient computation (5). Here, a Gaussian white noise signal $\sigma$ generated by a pseudo-random number generator (PRNG) in conjunction with an array of digital-to-analog converters (DAC) is added to the make value, and the break values are subtracted from the make values using differential amplifiers to calculate the gradient for each variable. Finally, a winner-takes-all (WTA) circuit identifies the variable with the highest gradient (6) and the output signal is used to update the register state using XOR gates (7).

Crucially, the relative simplicity of WalkSAT-XNF enables us to map every computational step to an equivalent analog circuit, enabling rapid continuous computation. As with other IMC concepts[8,34], the crossbar arrays in Fig. 2 enable parallel gradient computations for both the CNF and XOR clauses within a single clock cycle. Performing an entire operation of WalkSAT-XNF is achieved within just three clock cycles, without the need for a complex control system, while also circumventing frequent time-intensive communication with external co-processors or memory systems. Both XOR and CNF clauses can be evaluated using the same array, allowing for an area-efficient design. Moreover, the crossbar array can implement a number of literals per clause that is equal to the number of variables, hence supporting highly complex clauses common in industry workloads.

## Experimental demonstration using RRAM crossbar arrays

As with other mixed-signal computing systems, realizing WalkSAT-XNF in hardware requires it to be sufficiently resilient against hardware non-idealities in the analog circuits. Studies have identified variations in the RRAM cells and noise in the crossbar array's analog readout circuit as the dominant non-idealities that can result in a deterioration in performance[35]. To evaluate the feasibility of realizing WalkSAT-XNF in hardware, we implement a hybrid version of the architecture in Fig. 2 on an RRAM crossbar array chip. We experimentally validate the analog computation of clause evaluation and make/break value computation using an RRAM crossbar array chip, while the register, the circuits for checking clause satisfaction, the WTA circuit, and the Gaussian noise injection are emulated on a digital computer. The RRAM chip is a custom CMOS circuit in a 180 nm technology node with back-end-of-the-line (BEOL) monolithically integrated TaO$_x$ 1T1M RRAM cells[36,37]. For the experiment, we use the XNF instance derived from the par-8-1-c MDP problem[38], consisting of 13 variables and 42 clauses, including one XOR clause. To implement the crossbar's ON and OFF states $b_{ij}$, the RRAM cells are programmed to either a high-resistance state (HRS, or OFF state) or a low-resistance state (LRS, or ON state). Figure 3a shows the conductance values of the RRAM cells after programming. Here, the LRS is set to 100 µS and the HRS is set to 1 µS. Two separate arrays are used for the clause evaluation (array 1) and the make and break value computations (array 2). Figure 3b shows a histogram of the memristor conductances of array 1. The memristors exhibit typical device-to-device variations during programming[39], where the LRS and HRS are programmed to have a tolerance of ±10 µS. While further optimization is possible[40], we find that this accuracy is sufficient for our purposes.

To evaluate the capability of this crossbar array to perform clause evaluation (array 1 in Fig. 3a), we supply 400 random variable configurations as input signals and record the output current from the array. Figure 3c shows a histogram of the results, with distributions color-coded by the expected number of satisfied literals ($H$), showing a clear separation. It is thus possible to infer the number of satisfied literals directly from the array's analog output signal using the threshold levels indicated by the dotted lines in Fig. 3c with an average error of ~1%. The second array can be used similarly to evaluate the make and break values. We perform the make and break value computations sequentially here, but a parallel, pipelined evaluation is possible by employing two separate crossbar arrays. We then employ the gradient computation as part of the full WalkSAT-XNF heuristic. Figure 3d shows the cumulative success rate for solving par-8-1-c problem instance. We have performed 500 repeats at a noise level of $\sigma = 2.5$, where the solver runs for a maximum of 2000 iterations per repeat. The solver consistently finds a satisfying solution within this limit and experimental results align well with ideal (i.e., variation-free and noiseless) simulations despite hardware non-idealities.

We also compare experiments and simulations by varying the noise level $\sigma$. To quantify differences in the cumulative success rate, we analyze the iterations-to-solution (ITS$_{99opt}$). In Fig. 3e, we show ITS$_{99opt}$ for different noise levels and compare it against simulation-based results. Our results agree well with the experimental results, within the margin of error of the simulations. Overall, our results demonstrate that WalkSAT-XNF can be implemented using RRAM-based analog IMC hardware. The agreement between experiments and simulations highlights the robustness of the WalkSAT-XNF heuristic to hardware non-idealities, making it well-suited for implementation in custom CMOS circuits. This observation is also supported by a simulation-based sensitivity study, the results of which are presented in Supplementary Note 5. We believe this robustness to be due to the fact that the weights and the input states in our architecture are binary. The results of the crossbar array's operations are discrete integer values, thereby providing additional robustness against noise, compared to, for example, floating point operations.

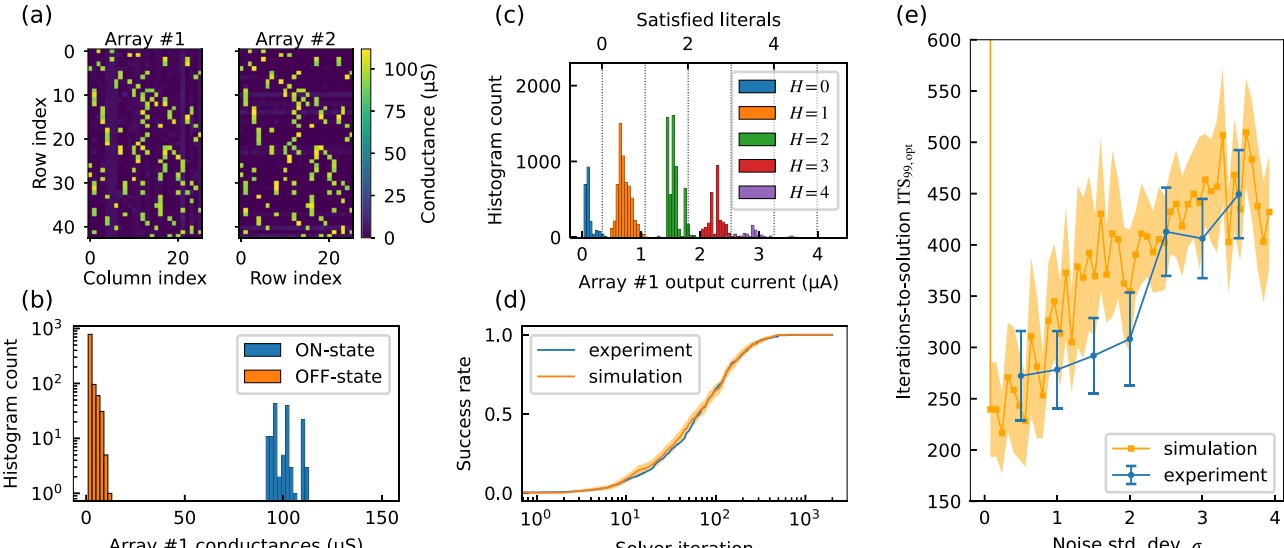

**Fig. 3 | Experimental demonstration of WalkSAT-XNF on TaO_x memristor crossbar arrays. a** Conductance map of the memristor crossbar arrays used for clause evaluation (array 1) and make and break value computations (array 2). **b** Histogram of the conductance values in array 1. **c** Histogram of the output currents of array 1 for 400 random variable assignments. The histogram is split and colored according to the expected number of true literals. Vertical lines indicate the discretization levels applied for clause evaluation. **d** Cumulative success rate when solving the par-8-1-c problem instance when implemented experimentally in the memristor crossbar arrays and simulations of WalkSAT-XNF heuristic presented in Table 1 for the noise $\sigma = 2.5$. **e** Comparison of iterations-to-solution values for different noise levels between experiments and simulations. Error bars for (**d**) and (**e**) depict the standard error[63].

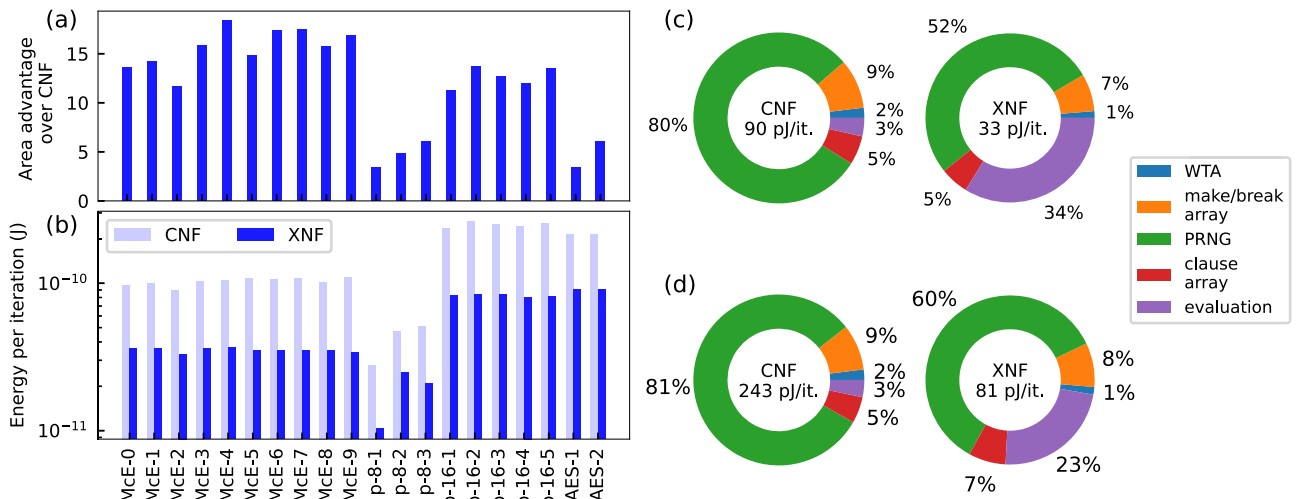

**Fig. 4 | Energy and area advantages of hybrid XOR–CNF formulations for in-memory hardware accelerators. a** Relative crossbar area between XNF and CNF benchmarking instances. **b** Average energy per iteration of WalkSAT-XNF for XNF and CNF benchmarking instances. **c, d** Relative contribution of the different hardware components to the energy consumption for the CNF and XNF representations of the benchmarking instances McEliece (**c**) and MDP (**d**).

## Simulation-based benchmarking for a 28 nm RRAM architecture

To evaluate our accelerator architecture illustrated in Fig. 2, we designed and simulated an architecture implementation using TaO_x RRAM crossbar arrays realized in a 28 nm CMOS process. For the simulations, we have derived latency and energy models from detailed circuit simulations and have evaluated them using activity simulations for the different SAT instances in Fig. 1. As our architecture supports both XOR and CNF clauses, we compare the CNF and XNF representations for the same problems on the same accelerator architecture to highlight the advantages for IMC accelerators of converting CNF instances to XNF instances. Figure 4a shows the average area advantage of XNF representations over CNF representations. We define the area advantage as $A_{XNF}/A_{CNF}$, where $A$ is the number of memory cells in

the crossbar arrays required for a given benchmarking instance. We find that XNF representations provide a $(12.2 \pm 4.7)\times$ average area advantage for the crossbar arrays due to there being a reduced number of variables and clauses. This significantly reduces the footprint, thereby enhancing the cost-effectiveness, scalability, and energy efficiency of the accelerator. Figure 4b shows the average energy per iteration of the WalkSAT-XNF heuristic. The median energy uptake for the XNF representation is 36 pJ (interquartile range (IQR): 47 pJ) compared to 107 pJ (IQR: 119 pJ) for the CNF representation, thereby achieving a ~3× improvement in energy efficiency. Figure 4c provides a breakdown of energy consumption across hardware components for a McEliece instance. For the CNF representation with 174 variables and 623 clauses, the average energy per iteration is ~90 pJ. Here, the

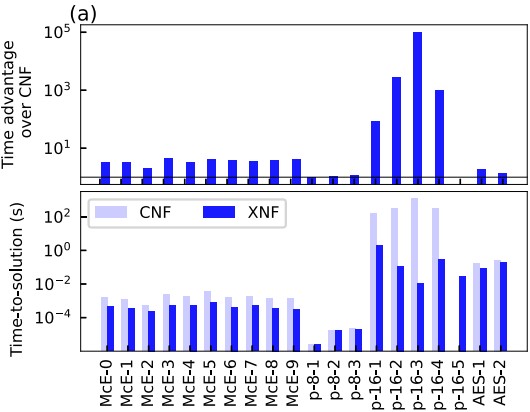
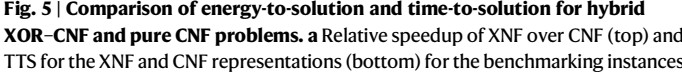
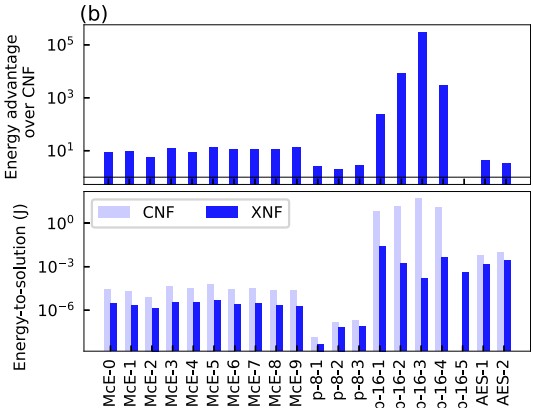

**Fig. 5 | Comparison of energy-to-solution and time-to-solution for hybrid XOR−CNF and pure CNF problems. a** Relative speedup of XNF over CNF (top) and TTS for the XNF and CNF representations (bottom) for the benchmarking instances using WalkSAT-XNF. **b** Relative energy advantage (top) and ETS for the XNF and CNF representations (bottom) for the benchmarking instances using WalkSAT-XNF. No data is shown for p-16-5, as no solution was found for the CNF representation.

majority of energy is consumed by the circuits responsible for generating the Gaussian noise signal (PRNG, ~80%), while the second-largest contributor (the clause evaluation array) accounts for only ~9% of the energy uptake. The make and break computation array, the evaluation circuits, and the WTA circuit combined contribute to ~10% of the energy consumption. For the XNF representation with 32 variables and 96 clauses (13 of which are XOR clauses), energy consumption drops to ~33 pJ, that is, only a third of the CNF instance. Moreover, we find that the relative energy contributions between the two representations are notably different as approximately a third of the energy consumption of the XNF representation is dedicated to the clause evaluation circuits. The XOR clause evaluation is energetically more expensive, which accounts for 93% of the energy uptake of the evaluation circuits. Figure 4d shows a comparison of this breakdown for a 16-bit MDP instance. The XNF representation shows lower relative energy consumption by the evaluation circuits compared to Fig. 4c, due to a lower XOR-to-CNF clause ratio (7% in the MDP instance versus 23% in the McEliece instance). Overall, while an XNF representation significantly reduces energy consumption, it introduces a trade-off: problem size reduction increases the number of XOR clauses which are more energy-intensive to evaluate.

Figure 5a shows the relative advantage of the time-to-solution (TTS) for the CNF and XNF representations. Here, the TTS is attained by multiplying $ITS_{99opt}$ with the latency of performing one iteration. We find that, in all instances, the TTS for the XNF instances is improved over the CNF representation with a median advantage of 3.7× (IQR: 22.2). Separated by instance classes, MDP instances show the greatest improvement (546×, IQR: 27,496.2), followed by McEliece (3.7×, IQR: 0.8) and AES (1.7×, IQR: 0.2). A further comparison between the CPU and hardware implementations of WalkSAT-XNF is provided in Supplementary Note 1, highlighting the additional speedups gained through IMC hardware acceleration.

To analyze the energy consumption of the accelerator architecture for the different problem representations, we consider the energy-to-solution (ETS). The ETS is calculated by multiplying $ITS_{99opt}$ with the average energy consumed per iteration. Figure 5b shows the relative ETS advantage of the XNF representation over the CNF representation. We find that energy consumption is improved over CNF with a median of 11.4× (IQR: 65.4). Separated by instance classes, we again observe that the MDP instances benefit most (1644.1×, IQR: 83540.7), followed by McEliece (11.4×, IQR: 3.4) and AES (3.9×, IQR: 0.6).

Beyond this comparison of different problem representations for IMC hardware accelerators, we benchmark our accelerator against SAT

solvers running on a CPU. For our benchmarking, the ETS and TTS were measured when running solvers on a 2.6 GHz Xeon CPU, and compared to the results for the XNF instances in Fig. 5. The TTS of the benchmarking solvers is directly derived from the CPU runtime. For the SAT solvers, we consider the SLS-solvers xnfSAT[16] and WalkSAT-SKC[30], alongside the conflict-driven clause learning (CDCL) solvers CryptoMiniSat[22] and Kissat[41]. The xnfSAT and CryptoMiniSat solvers are capable of solving problems in XNF representation and are therefore evaluated with XNF instances (see Supplementary Note 2 for more details). For xnfSAT, we initially noted that performance for pre-processed XNF instances is considerably worse compared to unprocessed XNF instances. To provide the fairest comparison, we therefore decided to evaluate the performance of xnfSAT using the unprocessed XNF instances, while WalkSAT-XNF and CryptoMiniSat were evaluated using the XNF-PP instances. WalkSAT-SKC and Kissat on the other hand support only CNF clauses and were therefore evaluated using the CNF representation of the benchmarking instances.

Figure 6 presents correlation plots comparing TTS and ETS for XNF-native solvers (a) and CNF-native solvers (b) against our WalkSAT-XNF accelerator. Table 2 summarizes the median relative performance. Compared to the best-performing software solver CryptoMiniSat, WalkSAT-XNF improves the median TTS by 9.1× and the ETS by $2.3 \cdot 10^3$×. Notably, while our accelerator outperforms CryptoMiniSat for the McEliece instances, most MDP and AES problems are solved faster by CryptoMiniSat. This indicates that the structure of such problems may be more favorable to CDCL-type solvers compared to the SLS heuristic employed in WalkSAT-XNF. However, WalkSAT-XNF demonstrates a smaller ETS in most instances compared to the CDCL-type solvers. We also note that, while WalkSAT-XNF is always able to find a solution, the SLS solvers xnfSAT and WalkSAT-SKC are unable to solve a portion of the MDP instances. Moreover, xnfSAT exhibits a large variance, while WalkSAT-XNF forms distinct clusters for similar class and size instances. This clustering pattern allows for a more stable prediction of performance of similar instances and can likely be attributed to the full-neighborhood evaluation, compared to xnfSAT's individual clause evaluation.

## Discussion
Our results show that IMC hardware accelerators for SAT problems can be enhanced to solve problems in a hybrid XOR−CNF representation, which is the native representation of several industrial optimization problems. By performing parallel gradient computation of XOR and CNF clauses on the same crossbar arrays, our approach enables a fast and energy-efficient hardware implementation of our WalkSAT-XNF

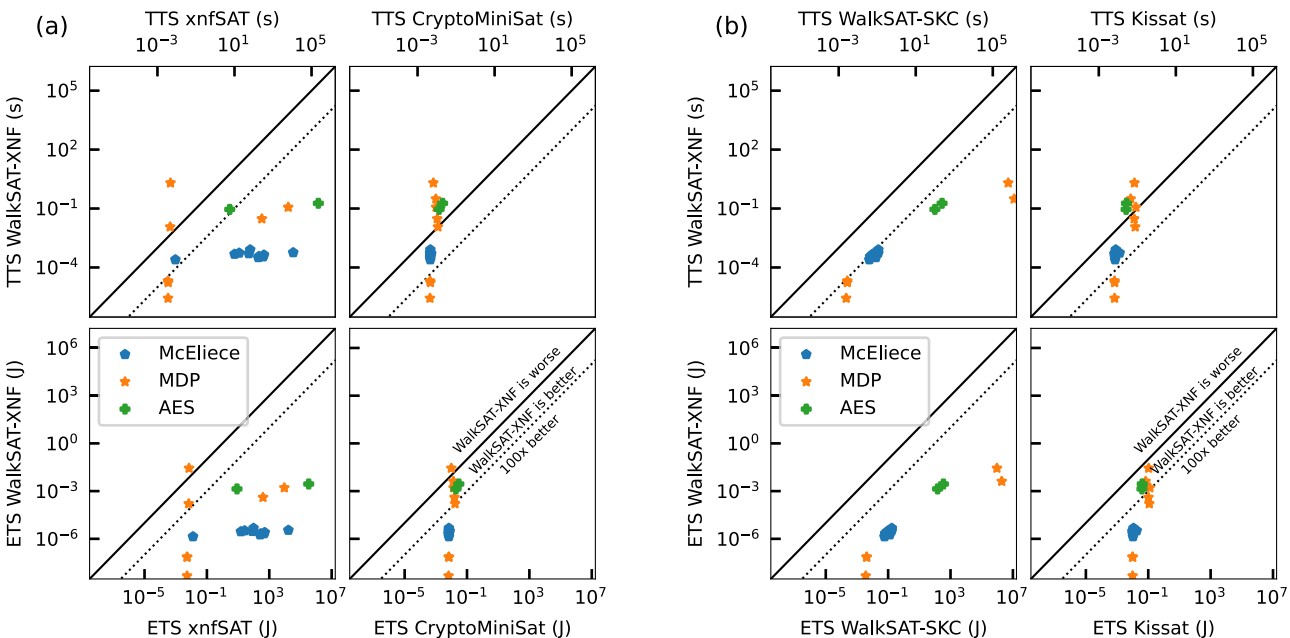

**Fig. 6 | Benchmark of energy-to-solution and time-to-solution against state-of-the-art SAT solvers. a** TTS and ETS benchmarking results comparing WalkSAT-XNF with the native XNF solvers xnfSAT and CryptoMiniSat. **b** TTS and ETS benchmarking results comparing WalkSAT-XNF to the CNF-native solvers WalkSAT-SKC and Kissat.

**Table 2 | Performance comparison of XOR–CNF and CNF solvers relative to WalkSAT-XNF**

| (a) | xnfSAT | | | CryptoMiniSat | | |
|---|---|---|---|---|---|---|
| | **Δ TTS** | **Δ ETS** | **Solved(%)** | **Δ TTS** | **Δ ETS** | **Solved(%)** |
| McEliece | $3.1 \cdot 10^5$ | $8.0 \cdot 10^7$ | 100 | 10.5 | $2.7 \cdot 10^3$ | 100 |
| | $(6.7 \cdot 10^5)$ | $(1.7 \cdot 10^8)$ | | (3.9) | (972.1) | |
| MDP | 758 | $5.3 \cdot 10^5$ | 87.5 | 0.7 | 72.5 | 100 |
| | $(2.0 \cdot 10^4)$ | $(2.2 \cdot 10^6)$ | | (229.5) | $(9.4 \cdot 10^4)$ | |
| AES | $6.1 \cdot 10^5$ | $6.0 \cdot 10^7$ | 100 | 0.13 | 12.5 | 100 |
| | $(6.1 \cdot 10^5)$ | $(6.0 \cdot 10^7)$ | | (0.02) | (1.8) | |
| all | $4.2 \cdot 10^4$ | $8.3 \cdot 10^8$ | 95 | 9.1 | $2.3 \cdot 10^3$ | 100 |
| | $(5.9 \cdot 10^5)$ | $(6.3 \cdot 10^{19})$ | | (13.0) | $(3.3 \cdot 10^3)$ | |
| (b) | WalkSAT-SKC | | | Kissat | | |
| | **Δ TTS** | **Δ ETS** | **Solved(%)** | **Δ TTS** | **Δ ETS** | **Solved(%)** |
| McEliece | 145.8 | $3.8 \cdot 10^4$ | 100 | 17.6 | $4.6 \cdot 10^3$ | 100 |
| | (51.6) | $(1.4 \cdot 10^4)$ | | (7.8) | $(2.0 \cdot 10^3)$ | |
| MDP | $2.2 \cdot 10^6$ | $2.5 \cdot 10^8$ | 62.5 | 4.5 | 488.3 | 100 |
| | $(1.5 \cdot 10^{18})$ | $(1.6 \cdot 10^{20})$ | | (334.5) | $(1.4 \cdot 10^5)$ | |
| AES | $1.2 \cdot 10^3$ | $1.1 \cdot 10^5$ | 100 | 0.2 | 21.6 | 100 |
| | (84.1) | $(8.3 \cdot 10^3)$ | | (0.1) | (7.0) | |
| all | 197.3 | $5.9 \cdot 10^4$ | 85 | 13.5 | $3.4 \cdot 10^3$ | 100 |
| | $(7.9 \cdot 10^4)$ | $(9.1 \cdot 10^6)$ | | (19.9) | $(5.4 \cdot 10^3)$ | |

Median time-to-solution (Δ TTS) and energy-to-solution (Δ ETS) relative to WalkSAT-XNF as well as percentage of solved instances for hybrid XOR–CNF solvers (a) xnfSAT and CryptoMiniSat and CNF solvers (b) WalkSAT-SKC and Kissat. The IQR is shown in brackets.

heuristic. This allows us to combine the algorithmic advantages of mapping problems to a hybrid XOR−CNF representation with the inherent parallelism and efficiency of IMC hardware.

For SAT problems that can be natively expressed as hybrid XOR−CNF problems, we find that this can reduce the chip area and energy consumption, while also improving the computation speed compared to mapping them to a pure CNF representation. This presents an advantage over existing SAT hardware accelerators, which can solve problems only in pure CNF formulation. When tackling pure CNF problems, the IMC architecture in Fig. 2 has

previously demonstrated that it can outperform comparable SAT accelerators (see Supplementary Note 3). As shown in our comparison in Fig. 4, the ability to implement XOR clauses can provide an additional order-of-magnitude improvement in computation speed and energy efficiency.

Moreover, the crossbar array embedding depicted in Fig. 2 can, in principle, support dense XOR and CNF clauses with as many literals as there are variables. Our experimental proof of concept successfully demonstrates this for a hybrid XOR−CNF problem with up to five literals per clause, which can be extended to even more complex clauses.

**Table 3 | Clause densities across problem representations**

| Class | CNF | | | CNF-PP | | | XNF | | | XNF-PP | | |
|---|---|---|---|---|---|---|---|---|---|---|---|---|
| | $k_{max}$ | $d_{CNF}$ | $d_{XOR}$ | $k_{max}$ | $d_{CNF}$ | $d_{XOR}$ | $k_{max}$ | $d_{CNF}$ | $d_{XOR}$ | $k_{max}$ | $d_{CNF}$ | $d_{XOR}$ |
| McEliece | 3 | 1.55 | – | 5 | 2.56 | – | 13 | 1.93 | 3.44 | 13 | 10.31 | 18.97 |
| MDP | 3 | 2.52 | – | 6 | 2.97 | – | 13 | 4.17 | 7.90 | 15 | 9.28 | 13.57 |
| AES | 5 | 0.82 | – | 5 | 0.97 | – | 14 | 1.43 | 3.57 | 12 | 3.25 | 5.16 |

Maximum number of literals per clause $k_{max}$ and average clause densities (in %) $d_{CNF/XOR}$ for CNF and XOR clauses for the different problem representations

This allows our architecture to additionally leverage the advantages of SAT preprocessing techniques, which tend to trade increased algorithmic efficiency with a higher density of literals per clause (see Table 3). By combining these advantages, we find that our proposed accelerator can outperform state-of-the-art SAT solvers running on digital computers in terms of computation speed and energy consumption.

As energy efficiency becomes an increasing concern in high-performance computing systems for resource-intensive applications such as optimization and artificial intelligence, hybrid XOR–CNF IMC accelerators can reduce operational costs and mitigate environmental impacts. In edge-computing applications, such as channel decoding in wireless receivers or AI route planning in autonomous vehicles, constraints on energy consumption and latency for computing hardware can benefit from fast and energy-efficient SAT accelerators to improve performance while enabling new use cases. Because XOR clauses are native to a wide variety of industry-relevant applications, such as hardware design, cryptanalysis, and telecommunications, we expect that a hybrid XOR–CNF SAT accelerator can provide considerable advantages when solving hard SAT problems.

While CNF and hybrid XOR–CNF instances have been identified as promising use cases for the IMC accelerator, there are also important industrial applications that rely on pure XORSAT problems. Although finding satisfying assignments to XORSAT problems is polynomial in problem complexity and thereby performed efficiently with linear system solvers on digital computers[6], there is a variety of hard industry-relevant XORSAT problems where the state-of-the-art heuristics rely on XORSAT evaluations, such as error correction[18] or efficiently attacking the McEliece cryptosystem[42]. For such problems, spin glass hardware accelerators have previously been demonstrated that scale exponentially in compute time[6,7] and it is likely that a native XOR–CNF accelerator can improve performance over existing techniques[43].

An interesting outcome of our research has been the insight that our proposed WalkSAT-XNF heuristic can benefit considerably from fast preprocessing techniques present in common SAT software libraries. By applying preprocessing to CNF instances before converting them to XNF instances, we have observed significant overall improvements in the number of iterations required to find a solution compared to XNF instances without preprocessing. While the hybrid XOR–CNF solver xnfSAT does not appear to benefit from preprocessing for the benchmarking instances we have studied, WalkSAT-XNF can improve the median TTS and ETS by an order of magnitude.

Although our results show there are clear advantages in using hybrid IMC XOR–CNF SAT accelerators, we envision possible improvements that could further enhance computational performance and relevance to industrial use cases. Our analysis of their energy consumption has identified the generation of noise signals and the evaluation of XOR clauses as targets for improvements. Enhancing the energy efficiency of noise signal generation would be possible by optimizing the PRNG design or by using analog noise sources[44]. Similarly, the circuit used for conducting parity checks could likely be improved, given that only the LSB is needed or that, alternatively, trees

of XOR gates can be employed. As we show in Supplementary Note 4, additional energy savings can also be achieved by reducing the resolution of the ADC.

One challenge in realizing performance enhancements for industrial applications pertains to the scalability of IMC hardware. Crossbar arrays are limited in size, for example, by parasitic effects, signal drop-off, and non-idealities, to a few hundred rows and columns. Current IMC hardware capable of dense matrix–vector operations could support the computations in our architecture for SAT problems with up to ~250 variables and ~500 clauses within a single array[45]. To overcome this limitation and increase the capacity for solving larger and more-complex SAT problems, one potential strategy would be to distribute the computational load by partitioning the variables and clauses across multiple crossbar arrays[33]. Exploring the implementation of such a multi-array architecture is an essential step in enhancing the scalability and applicability of our solver, opening up the possibility of solving larger and more-complex SAT instances.

The WalkSAT-XNF heuristic is an evolution of the CNF-specific WalkSAT heuristic and does not differentiate between XOR and CNF clauses for the purpose of variable selection. Based on the insights from this work, it could be possible to use IMC hardware for accelerating algorithmically efficient heuristics that include more sophisticated clause differentiation (e.g., by pre-solving the XOR clauses using Gauss–Jordan elimination[46]). Further enhancements can be achieved by combining it with the parallel tempering framework, which has recently been shown to provide performance improvements for IMC architectures with minimal overhead[47]. Finally, high-performance SAT solvers often combine CDCL and SLS heuristics, including XOR subroutines[48,49]; our IMC approach could similarly be adapted to accelerate other types of heuristics, including CDCL SAT solvers[50].

## Methods
### Benchmarking instances
**McEliece–Niederreiter cryptosystem.** The McEliece instances are derived from cryptographic attacks[25,51] on the McEliece–Niederreiter cryptosystem[52,53]. This cryptosystem was proposed as the first code-based public-key cryptosystem in the 1970s and has been elected by the National Institute of Standards and Technology (NIST) as a quantum-resistant public-key cryptographic algorithm for evaluating post-quantum cybersecurity[54].

For the encryption and decryption of a cipher, the receiver generates three matrices: the $n$-by-$k$ generator matrix $G$ typically using Goppa codes; an $n$-by-$n$ permutation matrix $P$; and a random $k$-by-$k$ invertible matrix $S$. The receiver publishes a public key $G' := SGP$. The message sender prepares a plaintext message $m$ and creates the ciphertext $y = m^T G' + e$, where $e$ is an error vector with a Hamming weight of $t$. The receiver then uses an error-correction algorithm[55] to identify the error vector $e$ and obtains $m$ via $G$, $P$, and $S$. A potential attack on the McEliece cryptosystem involves identifying the error vector $e$. In particular, the authors in ref. 25 interpret the problem as finding the minimum-weight codeword. Let $H$ be an $(n - k) \times n$ matrix, with $H_{i,j}$ being the $(i, j)$-th element of the matrix $H$. The linear system $Hc = 0$ is then written over the binary field with the XOR logical

operator $\oplus$. For instance, the $i$-th equality of $Hc = 0$ is

$$H_{i,1}c_1 \oplus H_{i,2}c_2 \oplus \cdots \oplus H_{i,n}c_n = 0. \quad (4)$$

A decoding attack on the system involves finding a solution $c$ to $Hc = 0$ having the desired Hamming weight.

Based on this attack, the McEliece instances are generated via the PySA package[56] (further details can be found in refs. [26,57]). Each instance is first generated as a set of XOR equations as shown in Eq. (4). The XOR equations are then translated to CNF clauses, and the Hamming weight of the desired solution $c$ is incorporated using additional CNF clauses. We use 10 CNF instances with a code length equal to 16. We label these instances from McE-$i$, where $i \in \{0, ..., 9\}$. The numbers of variables and clauses range from 171 to 183, and 611 to 659, respectively.

**Minimal disagreement parity problem.** The MDP instances are generated from the minimal disagreement parity problem described in ref. [38]. Given an $m$-by-$n$ binary matrix $X$, a binary vector $y$ of length $m$, and an integer $k$, the MDP problem seeks to find a binary vector $a \in \{0, 1\}^n$ satisfying

$$\sum_{i=1}^{m} \left( \left( \sum_{j=1}^{n} X_{i,j} a_j \right) \oplus y_i \right) \leq k. \quad (5)$$

The difficulty in solving the MDP problem has been explored in the literature, and an algorithm for solving the inequality (5), relying on XOR clauses only, was suggested in ref. [58]. A total of 15 MDP instances were proposed by Crawford[38] and added to the DIMACS library[59], with the instances translated to a CNF representation. We selected 10 instances, par-8-$i$-c and par-16-$i$-c, $i \in \{1, ..., 5\}$, from the DIMACS library[59], and they can be accessed from ref. [57]. We labeled these instances p-8-$i$, p-16-$i$, where $i \in \{1, ..., 5\}$. The numbers of variables and clauses lie in the ranges [64, 74] and [254, 298] for the par-8-$i$-c family, and [317, 349] and [1264, 1392] for the par-16-$i$-c family.

**Advanced encryption standard.** The Advanced Encryption Standard (AES)[27,28] is a symmetric key encryption algorithm selected by the National Institute of Standards and Technology (NIST). It was developed to replace an older data encryption standard (DES) that was shown to be vulnerable to decryption attacks, particularly with the advent of stronger computational resources. Applications of AES include securing communications for online financial transactions and encrypting data in a database[60]. XOR operations are one of the key components of the encryption process that utilizes the so-called round keys, which are inherent to AES and finding them is indicative of a successful cryptographic attack. Instances pertaining to AES are available in the dataset from the 2012 SAT competition[61], and they can be accessed from ref. [57]. Solving these problem instances is viewed as

a successful cryptographic attack to AES. As mentioned in ref. [61], these instances inherit XOR operations, but are translated into a CNF representation, making it possible to utilize SAT solvers that operate only CNF clauses. We use instances called aes_32_1_keyfind_$i$, where $i = 1, 2$ and label them AES-1 and AES-2 in the benchmarking experiment below. The numbers of variables and clauses are 300 and 1056, respectively.

## XNF problem conversion

We provide the details on the conversion process for generating the formulation classes CNF-PP, XNF, and XNF-PP, which illustrated in Fig. 1b. We incorporated CNF preprocessing using PySAT[62], a Python library designed to work with SAT instances with CNF clauses only. We use PySAT to access the CaDiCaL solver's preprocessor[29]. To produce preprocessed CNF instances (denoted by CNF-PP in the figure), the parameter named 'rounds' was set to 3, indicating the number of preprocessing rounds. PySAT supports a variety of preprocessing techniques, including blocked clause elimination, covered clause elimination, globally blocked clause elimination, equivalent literal substitution, bounded variable elimination, failed literal probing, hyper binary resolution, clause subsumption, and clause vivification. Details on each technique can be found in ref. [29]. All available preprocessing techniques supported by the package were employed, provided by the following parameters: block, cover, condition, decompose, elim, probe, probehbr, subsume, and vivify. The time to process a CNF instance to its preprocessed counterpart CNF-PP ranges approximately from $2 \cdot 10^{-3}$ to $1 \cdot 10^{-2}$ s, with an average time of around $7 \cdot 10^{-3}$ s.

To convert an instance in CNF representation into XNF form, we employed the cnf2xnf tool, which is a utility present in the xnfSAT solver[16]. The cnf2xnf tool is designed to transform CNF instances by identifying and extracting XOR clauses from given CNF clauses. The resulting hybrid representation retains the structure of the original CNF instance while introducing XOR clauses, making the clauses more compact. The processing time to convert a CNF instance to an XNF instance ranges from $-3 \cdot 10^{-3}$ to $3 \cdot 10^{-2}$ s, with an average time of around $4 \cdot 10^{-3}$ s. For converting a CNF instance to XNF form, the processing time ranges from $2 \cdot 10^{-3}$ to $4 \cdot 10^{-3}$ s, with an average time of around $3 \cdot 10^{-3}$ s.

Table 3 presents the average of clause densities of each instance class, where the density is calculated by summing the number of literals in each clause and dividing by the total number of variables. We present further observations regarding the literals per clause densities $d_{CNF}$ and $d_{XOR}$ of the XNF and XNF-PP formulation classes in Supplementary Note 1.

## Benchmarking of SAT solvers on CPUs

The TTS and ETS of xnfSAT, CryptoMiniSat, WalkSAT-SKC, and Kissat were calculated using an Intel Xeon CPU running at 2.60 GHz with

**Table 4 | Solver parameters and energy-to-solution (ETS) estimation methodology**

| (a) Noise parameter for WalkSAT-XNF | | | | | |
|---|---|---|---|---|---|
| Algorithm | Parameter | Formulation | Instance | | |
| | | | McEliece | MDP | AES |
| WalkSAT-XNF | Noise ($\sigma$) | CNF | 2.5 | 2.5 | 1.0 |
| | | XNF | 3.0 | 2.5 | 1.5 |
| **(b) ETS estimates used for solvers** | | | | | |
| Solvers | | | ETS Estimate | | |
| WalkSAT-XNF | | | Average joules per iteration × ITS | | |
| CryptoMiniSat, Kissat, WalkSAT-SKC | | | 1.5 watts × TTS | | |

(a) Noise parameter ($\sigma$) used for WalkSAT-XNF across different problem formulations and instances. (b) Calculation methodology for energy-to-solution (ETS) estimates for benchmarking solvers

512 GB of system memory and 128 virtual cores. For the ITS and TTS estimations, the number of trials was set to 1000 by all algorithms and instances in order to obtain a reliable success probability $\theta$[63]. For CryptoMiniSat, the parameter named 'maxsol' was set to 1, quantifying the number of targeted solutions found by the algorithm. The maximum allowed runtime for Kissat was set to 300 s. For WalkSAT-XNF, WalkSAT-SKC, and xnfSAT, each trial was capped at $10^9$ maximum allowed bit flips. The noise parameters used for WalkSAT-XNF were optimized in a grid search for the different problem classes. The optimized parameters are displayed in Table 4a. The computation of the ETS for each solver is outlined in Table 4b.

To estimate the energy consumption of solvers that solely depend on software, 1.5 joules per second (i.e., 1.5 watts) was used. We benchmarked several instances using CryptoMiniSat on an AMD Epyc server while tracking the energy usage using the Powertop package[64]. In all cases, we observed 1.5 watts, which we used as the baseline energy usage for all CPU-based solvers. Of note, the full benchmarking experiments were performed on Intel Xeon CPUs running at 2.80 GHz with 90 GB of RAM and 64 logical cores on the Google Cloud Platform (GCP), on which it is not possible to measure the energy directly. We believe our estimate of 1.5 watts is conservative, as the per-core thermal design can have a higher power ceiling.

### Hardware accelerator energy modeling

The components of the hardware architecture in Fig. 2 have been designed, validated, and modeled in a TSMC 28 nm technology node. The crossbar array is modeled for a BEOL integrated RRAM device using TaO$_x$ memristors based on data from previously fabricated test chips[37]. The output currents at the bit lines are detected and processed using transimpedance amplifiers with active common-drain feedback. For CNF clauses, output signals are evaluated with comparators based on a StrongARM latch architecture. For the XOR clause evaluation, we model the energy consumption of the ADCs based on a regression analysis of the ADC survey data in refs. 65,66. Based on the maximum number of literals for the benchmarking problems (see Table 3), we assume an ADC bit resolution of 4 bits, which can support clauses with up to 15 literals. For an ADC with a sampling rate of 900 million samples per second and a bit resolution of 4 bits, we estimate an energy consumption per operation of 0.718 pJ and an area of $3.9 \cdot 10^{-3}$ mm$^2$.

The Gaussian noise signal is generated from an XORSHIFT-64 PRNG using the Alias method. The normal-distributed random number sequence generated by the PRNG is converted to analog signals using R2R ladder DACs at each bit line of the gradient evaluation crossbar ((4) in Fig. 2). The WTA circuit is realized using voltage-controlled delay lines, whose output is evaluated using merger trees and arbiters. The one-hot encoded output of the WTA circuit is fed into an array of XOR gates, whose other input is the current variable configuration stored in the register. The output is used to set the new state of the register.

The circuit is driven and synchronized by a central clock signal, where the signal provided by the register sequentially progresses through the individual circuit blocks shown in Fig. 2. A single iteration of WalkSAT-XNF is performed in three clock cycles. During the first clock cycle, the signals are applied to the first crossbar array and the output signals are analyzed using the readout circuit. During the second clock cycle, the second crossbar array is operated in the same way. During the third clock cycle, the WTA operation is performed, and the register state is updated. The combined latency of these components per iteration of WalkSAT-XNF was modeled as taking $t_{iter} = 6$ ns. Once the register is initialized, the entire circuit will continuously repeat this flow until a predefined number of iterations is reached or until a satisfying solution has been identified. Additional details about the circuit designs and the hardware parameters can be found in ref. 8.

From these modeling results, a semi-analytical model has been derived, which evaluates the energy consumption of the individual components based on average signal levels and activity patterns. For the benchmarking, we have built a custom cycle-accurate simulator that derives instance-specific activity patterns and signal levels when running the WalkSAT-XNF heuristic. Using the semi-analytical model, we derive the mean energy consumption for each instance without the need for extensive SPICE-like simulations, which would be intractable. We derived the mean energy consumption per iteration of the WalkSAT-XNF heuristic $E_{mean/iter}$ for each instance and calculated the energy to solution as ETS = $E_{mean/iter} \cdot$ ITS.

### Experimental validation of the WalkSAT-XNF heuristic on memristor crossbar arrays

The experimental setup used to realize our IMC architecture comprises a custom chip fabricated in a TSMC 180 nm technology node and houses three 64-by-64 memristor crossbar arrays. The 1T1M cells are based on Ta/TaO$_x$/Pt RRAM that was monolithically integrated in-house in a BEOL process. To perform in-memory computations, the chip contains digital control and analog sensing circuits. Input signals to each array's word line are applied digitally and the analog output is reconstructed using the 'shift and add' method[67]. To convert and measure the signals from the array's bit lines, transimpedance amplifiers and sample-and-hold circuits are employed that rapidly convert the output currents to voltage signals and sample them. The signals are then converted to digital signals using ADCs. The chip is hosted on a custom-printed circuit board, which facilitates the voltage supply to the chip and provides a digital interface to access, control, and program the individual crossbar arrays. Additional details about the layout and the fabrication of the chip may be found in ref. 36. For the implementation of the WalkSAT-XNF heuristic, a custom Python program was written that performs the matrix operations in Fig. 2 on the crossbar arrays. Here, the matrices in Fig. 3a were programmed into two of the chip's arrays. During the matrix operations, the binary input signals are communicated to the chip and the output signals are measured and returned via the digital interface. For the clause evaluation, the number of true literals is inferred from the output signal using equidistant quantization levels. These levels have been optimized to yield the lowest error rate.

## Data availability

The benchmarking instances used in this study are available in ref. 57.

## Code availability

The simulator used for the heuristic simulation and energy modeling is open-sourced and available at https://github.com/HewlettPackard/CountryCrab.

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

## Acknowledgements

The authors thank our editor, Marko Bucyk, for his careful review and editing of the manuscript, and Dmitri Strukov for discussions on XOR hardware architectures. This material is based upon work supported by the Defense Advanced Research Projects Agency (DARPA) through Air Force Research Laboratory Agreement No. FA8650-23-3-7313. The views, opinions, and/or findings expressed are those of the author(s) and should not be interpreted as representing the official views or policies of the Department of Defense or the U.S. Government.

## Author contributions

H.I. and F.B. contributed equally to this work and are recognized co-first authors. H.I. and F.B. wrote the manuscript. H.I., N.K., and T.B. performed algorithm designs. M.N. and E.V. analyzed the numeric results. H.I., X.Z., and C.-W.Y. conducted the corresponding numeric benchmarking simulation. A.H. performed circuit and architectural simulations. X.S., J.I., and J.P.S. contributed to the memristor fabrication and experimental system development. G.P. and T.V.V. conceived the idea of asserting XOR clauses with in-memory computing. F.B. derived the hardware architecture, conducted the hardware modeling and energy simulations, and performed the hardware experiments. I.R. conceived the main idea of the XOR–CNF use case. I.R., T.V.V., J.P.S., M.M., and R.B. supervised and led the collaboration effort. All authors analyzed and discussed the results.

## Competing interests

The authors declare no competing interests.
