## [Transparent Peer Review file · Nature Communications]

Accelerating Hybrid XOR-CNF SAT Problems Natively with In-Memory Computing

Corresponding Author: Dr Ignacio Rozada

Version 0:

Reviewer comments:

Reviewer #1

(Remarks to the Author)

This paper proposes an in-memory computing architecture that implements and solves hybrid XOR-CNF (XNF) problems. A WalkSAT-XNF heuristic and a corresponding RRAM-based in-memory computing circuits are proposed to improve the speed and energy efficiency. The key novelty stems from providing XNF-native support through the algorithm-hardware co-design. The authors demonstrate a proof-of-concept implementation using fabricated RRAM arrays, and simulations show improved performance over CPU-based SAT solvers. However, the evaluation lacks comprehensiveness, and comparisons with state-of-the-art works are insufficiently addressed.

- The paper does not sufficiently introduce the state-of-the-art works in XNF solvers. Without this context, it is difficult to assess the novelty of WalkSAT-XNF. The authors are encouraged to review and discuss existing XNF solvers, their performance bottlenecks, and how the proposed approach addresses these limitations.

- The challenges of implementing the proposed architecture for XNF problems are unclear. While the concept of using in-memory computing for SAT solving is not new, the specific challenges in extending it to support hybrid XOR-CNF formulations are not clearly articulated.

- The conversion from CNF to XNF is confusing. The authors claimed that translating XOR clauses to CNF clauses is computationally expensive, but the benchmarks in this work seem to be implemented initially in CNF clauses and require conversion to XNF. Is this procedure also costly? Besides, for an XNF problem, is it required to be converted to CNF for preprocessing and afterwards converted back to XNF? A clear explanation would be appreciated.

- A major weakness of this work is the lack of comparison with the state-of-the-art works, making the improvement of this work less convincing. In addition to intrinsic comparison between XNF and CNF, the proposed architecture is only benchmarked with CPU implementations.

What is the improvement of the proposed in-memory computing architecture over existing in-memory computing works for SAT problems? Some references are listed below:

[1] D. Kim, N. M. Rahman and S. Mukhopadhyay, "29.1 A 32.5mW Mixed-Signal Processing-in-Memory-Based k-SAT Solver in 65nm CMOS with 74.0% Solvability for 30-Variable 126-Clause 3-SAT Problems," 2023 IEEE International Solid-State Circuits Conference (ISSCC), San Francisco, CA, USA, 2023, pp. 28-30.

[2] C. Shim, J. Bae and B. Kim, "30.3 VIP-Sat: A Boolean Satisfiability Solver Featuring 5x12 Variable In-Memory Processing Elements with 98% Solvability for 50-Variables 218-Clauses 3-SAT Problems," 2024 IEEE International Solid-State Circuits Conference (ISSCC), San Francisco, CA, USA, 2024, pp. 486-488.

What is the improvement of the proposed WalkSAT-XNF algorithm over existing CNF and XNF solvers if both implemented on CPUs?

-The evaluation of the detailed improvement of in-memory architecture is also missing, which makes the necessity of using in-memory computing unclear. Please provide the improvement of the proposed architecture compared with implementing WalkSAT-XNF on CPUs.

- What is the maximum problem size (in terms of variables and clauses) that the proposed RRAM crossbar arrays can support considering device non-idealities?

- What is the solvability performance of the proposed algorithm and architecture?

Reviewer #2

(Remarks to the Author)

In this paper, the authors introduced a WalkSAT-XNF, a stochastic local search (SLS) heuristic designed to natively handle hybrid XOR–CNF Boolean formulas. The key idea is to extend the classical WalkSAT framework to directly evaluate both CNF and XOR clauses without requiring conversion into pure CNF form. The authors further demonstrate how this algorithm can be efficiently implemented using analog crossbar arrays based on memristive devices, leveraging their parallelism for gradient computation in variable selection. However, my major concern is that the experimental evaluation lacks sufficient comparison with modern XOR-CNF SAT solvers (aaai.v33i01.33011592)(978-3-030-53288-8_22)(s11786-024-00594-x), especially those that already support native XOR clause handling. The lack of relevant comparisons also worries me about the innovation level of the algorithm. Secondly, there is also a lack of comparison with existing IMC-based solvers (ISSCC42615.2023.10067380) in terms of hardware; only comparing with CPU-based solvers is limited.

Below are the suggestions:

1. The paper primarily compares WalkSAT-XNF with traditional CNF-only solvers (e.g., Kissat, WalkSAT-SKC), which is insufficient. A meaningful comparison should include Advanced XOR-SAT solvers (aaai.v33i01.33011592 / 978-3-030-53288-8_22 / s11786-024-00594-x), specifically designed to handle CNF-XOR formulas arising from hashing-based approximate model counting. Without such comparisons, it is unclear whether the observed performance improvements are due to the IMC architecture or simply because the problems are better suited to XOR-enhanced formulations.
2. The paper claims that WalkSAT-XNF is uniquely suitable for IMC acceleration, but does not clearly explain why other algorithms (e.g., CDCL + Gauss-Jordan elimination) cannot be adapted to similar architectures.
3. No discussion is provided regarding why certain problem classes benefit more from one approach over another, nor is there a breakdown of structural features (e.g., XOR clause density, clause length, variable dependencies) that influence solver performance.
4. The paper emphasizes the acceleration and energy efficiency advantages of the IMC architecture over traditional CPUs, but does not compare with other IMC-based SAT accelerators such as Snap-SAT (ISSCC42615.2023.10067380).
5. Figure 2 of the paper illustrates the hardware module partitioning of WalkSAT-XNF, including clause evaluation, make/break computation, WTA, and other components, but provides no detailed description of its control logic or synchronization mechanisms. How do these modules coordinate with each other? Is there any control bottleneck?
6. WalkSAT-XNF utilizes analog signals to perform parallel evaluation of make/break values, but does not quantify the precision loss of analog components such as ADCs and comparators. What is the impact of different ADC resolutions on ITS (Iterations-to-Solution)?

Version 1:

Reviewer comments:

Reviewer #1

(Remarks to the Author)

The authors have addressed most of my questions, but a few concerns remain not well resolved.

- The maximum problem size that the proposed analog CiM array can support is limited primarily by the variations and on/off ratios of RRAMs as well as the non-idealities of the ADCs, instead of by the RRAM array size. Therefore, the claim of supporting up to 250 variables and 500 clauses is not convincingly substantiated.

- The comparisons with the state-of-the-art CiM-based SAT accelerators are not apple-to-apple. In Supplementary Table 3, as the authors have mentioned, the proposed work uses a more advanced technology node, and thus the results are not directly comparable. The source of the improvement on pure CNF problems against other baselines is also unclear. Besides, the authors claimed “our experimental system reported in [8] is a fairer base of comparison”, yet [8] is not compared. The use of [10] to cite the proposed work within the table is also very confusing.

Reviewer #2

(Remarks to the Author)

The authors have provided detailed responses to the initial reviews and have made substantial revisions to the manuscript. However, several points still require clarification or further analysis to strengthen the paper:

1. Since CDCL+GJE could also be accelerated via IMC, please elaborate on why WalkSAT-XNF was chosen as the first target instead of other algorithms. Give a brief discussion of the relative feasibility and potential performance of IMC-accelerated CDCL+GJE.
2. The comparison table shows that other accelerators do not natively support XNF. However, is it not possible to pre-process the XNF problems into a low-literal CNF formulation to run on these accelerators? If so, what would be the expected

performance penalty, and how would it affect the overall advantage claimed for your architecture?

3. For larger crossbar arrays, could the clock tree distribution and signal skew become a challenge for synchronization? Has this been analyzed or simulated?

4. The simulation results are promising, but the reliance on idealistic models (28nm, no leakage) may overstate the advantages. Please provide a sensitivity analysis of key non-idealities (e.g., conductance variations, ADC noise) on the solver's success rate and energy efficiency.

Version 2:

Reviewer comments:

Reviewer #1

(Remarks to the Author)

The authors have made substantial improvements in the discussion of hardware non-idealities as well as in the comparison with the state-of-the-art works, and have addressed my previous concerns. One remaining issue is the reference numbering. The authors note that the references in the response letter are as numbered in the revised manuscript and the Supplementary Information. However, I found that references [8] and [14] in the response letter correspond to [10] and [11] in the revised Supplementary Information. This inconsistency confuses me. Please make sure that the reference numbering is consistent and correct across the main manuscript and the Supplementary Information.

Reviewer #2

(Remarks to the Author)

The author provided detailed responses to the previous suggestions. After review, I believe that the authors have adequately addressed the issues raised and my previous concerns have been resolved. I suggest accepting the current form of the manuscript.

REVIEWER COMMENTS

Reviewer #1 (Remarks to the Author):

This paper proposes an in-memory computing architecture that implements and solves hybrid XOR-CNF (XNF) problems. A WalkSAT-XNF heuristic and a corresponding RRAM-based in-memory computing circuits are proposed to improve the speed and energy efficiency. The key novelty stems from providing XNF-native support through the algorithm-hardware co-design. The authors demonstrate a proof-of-concept implementation using fabricated RRAM arrays, and simulations show improved performance over CPU-based SAT solvers. However, the evaluation lacks comprehensiveness, and comparisons with state-of-the-art works are insufficiently addressed.

-> We would like to thank the reviewer for taking the time to read our manuscript and for the thoughtful questions. We appreciate the opportunity to clarify and have provided direct responses to each comment below.

1. The paper does not sufficiently introduce the state-of-the-art works in XNF solvers. Without this context, it is difficult to assess the novelty of WalkSAT-XNF. The authors are encouraged to review and discuss existing XNF solvers, their performance bottlenecks, and how the proposed approach addresses these limitations.

-> We thank the referee for pointing out the need to better position our work within the landscape of existing XNF solvers.

In the revised manuscript, we have incorporated a more detailed discussion of prior work on CryptoMiniSat and xnfSAT, which we believe represent the current state-of-the-art XNF solvers. This view is supported by recent work such as 'SAT Solving Using XOR-OR-AND Normal Forms, Mathematics in Computer Science, 18, 1 (2024) by Andraschko et al.' published last year, which identifies these two solvers as leading approaches for handling XOR+CNF instances. We used xnfSAT as a baseline for stochastic-local-search algorithm and CryptoMiniSat as a baseline solver following the CDCL-paradigm.

WalkSAT-XNF differs significantly in methodology from the CPU-based solvers. To the best of our knowledge, there are few, if any, published works that adapt stochastic-local-search algorithms to natively handle XOR constraints using 'IMC hardware'. This distinction is

important, as IMC architectures impose constraints that are fundamentally different from those of CPU-based implementations.

We provide detailed replies within the second question below to address the novelty of our approach. We also added Section 2 in the Supplementary Information to include discussions of existing benchmarking of XNF solvers along with their performance bottlenecks; the application of CryptoMiniSat; a broader class of instances called XOR-OR-AND normal form, which can be solved using 2-Xornado and WDSat, and the challenges that arise when converting instances into a formulation suitable for CNF+XOR solvers.

2. The challenges of implementing the proposed architecture for XNF problems are unclear. While the concept of using in-memory computing for SAT solving is not new, the specific challenges in extending it to support hybrid XOR-CNF formulations are not clearly articulated.

-> We agree with the referee that we should do a better job highlighting the challenges and the novelty of our approach, compared to in-memory accelerators for solving SAT problems in CNF formulation. For solving XNF problems with hardware accelerators, we observe two major challenges:

(i) Compared to CNF problems, XNF problems require considerably more complex hardware for clause evaluation. To evaluate CNF clauses, it is only necessary to check whether at least one literal is true. In our in-memory architecture, this can easily be performed with a single comparator. For XOR clause evaluation on the other hand, we need to know the number of true literals and the associated parity. In our in-memory approach, this requires the ability to obtain the Hamming distance from the crossbar array output with low error. It also requires an additional circuit that can perform the parity check using the analog output signal. It was initially not obvious whether the overhead of this more complex evaluation circuit offsets the higher algorithmic efficiency of converting SAT problems to XNF. To our knowledge, our work is the first that demonstrates an advantage for hardware accelerators in solving XNF problems instead of equivalent CNF problems, even when taking into account the additional overhead required for XOR evaluations.

(ii) XNF problems can require support for many literals. While the original CNF versions of our benchmark problems do not have more than 3 literals per clause, the XNF problems can have up to 15 literals per clause. In hardware accelerators, we observe that implementing such dense clauses can be challenging, as high clause densities can require more circuit elements to perform the clause evaluation. For example, with a binary tree of OR gates, the number of logic gates required for clause evaluation grows linearly with the number of literals. We thus find that most hardware accelerators reported in the literature are limited to ~ 3 literals per clause. Our work shows that supporting a high number of literals in XNF clauses using in-memory computing is feasible and can be energy-efficient enough to show a notable advantage over SAT accelerators that implement the problems using low-density CNF clauses.

In response, we have modified the Introduction section:

“Combining the advantages of a hybrid XOR--CNF formulation with IMC hardware could offer considerable advantages in tackling computationally challenging SAT problems with inherent XOR clauses. However, compared to pure CNF problems, evaluating XOR clauses requires more complex and energy-intensive circuits that can potentially offset the efficiency and latency advantages of IMC hardware. Moreover, XOR clauses can contain many literals, whereas SAT hardware accelerators can often only support few literals per clause. For IMC hardware, the high number of literals can also make it more challenging to retain low error rates during computation, as analog signals will have an increased dynamic range. It is thus still an open question whether IMC is suitable to efficiently accelerate hybrid CNF--XNF problems. Therefore, in this work we set to address the open question of whether IMC is suitable to efficiently accelerate hybrid CNF--XNF problems.”

We have also extended TABLE 2 in Section 4.2 to show the maximum number of literals per clause required to embed the SAT benchmark problems in CNF and XNF representation.

3. The conversion from CNF to XNF is confusing. The authors claimed that translating XOR clauses to CNF clauses is computationally expensive, but the benchmarks in this work seem to be implemented initially in CNF clauses and require conversion to XNF. Is this procedure also costly? Besides, for an XNF problem, is it required to be converted to CNF for preprocessing and afterwards converted back to XNF? A clear explanation would be appreciated.

-> We thank the referee for highlighting the need to improve this section in the manuscript. Direct encoding of XOR clauses into CNF clauses is costly, particularly when the XOR

clauses contain many literals, as it leads to an exponential number of CNF clauses. This exponential blow-up is illustrated in equation (2) of the manuscript. This is one of the key motivations behind using a native representation for the XOR constraints. We argue that solving an instance in its CNF form typically requires more computational resources, as it introduces more variables and clauses compared to its XNF counterpart. This observation is illustrated in FIG. 5 in Section 2.5 and FIG. 1 in the Supplementary materials.

In our experiments, all benchmark instances were originally sourced in the pure CNF formulation, as stated in the sentence “All instances inherit native XOR clauses but are initially provided with CNF clauses only.” in the manuscript. Our goal is to demonstrate that when these instances are reformulated as XOR+CNF, the resulting formulation is not only more compact but also more amenable for efficient solving by XOR-aware SAT solvers. The conversion from CNF to XOR+CNF is not observed to be computationally costly. As reported in Section 4.2 of the manuscript, the conversion time typically ranges from $O(1e-3)$ to $O(1e-2)$ seconds per instance.

We have added “Details of the preprocessing procedure and the per-instance preprocessing runtime are reported in Section 4.” Section 2.1 of the manuscript to direct readers where details can be found regarding the preprocessing steps.

4. A major weakness of this work is the lack of comparison with the state-of-the-art works, making the improvement of this work less convincing. In addition to intrinsic comparison between XNF and CNF, the proposed architecture is only benchmarked with CPU implementations.

* What is the improvement of the proposed in-memory computing architecture over existing in-memory computing works for SAT problems? Some references are listed below:

[1] D. Kim, N. M. Rahman and S. Mukhopadhyay, "29.1 A 32.5mW Mixed-Signal Processing-in-Memory-Based k-SAT Solver in 65nm CMOS with 74.0% Solvability for 30-Variable 126-Clause 3-SAT Problems," 2023 IEEE International Solid-State Circuits Conference (ISSCC), San Francisco, CA, USA, 2023, pp. 28-30.

[2] C. Shim, J. Bae and B. Kim, "30.3 VIP-Sat: A Boolean Satisfiability Solver Featuring 5×12 Variable In-Memory Processing Elements with 98% Solvability for 50-Variables 218-Clauses 3-SAT Problems," 2024 IEEE International Solid-State Circuits Conference (ISSCC), San Francisco, CA, USA, 2024, pp. 486-488.

-> We agree with the referees that additional comparisons against other state-of-the-art SAT hardware accelerators are helpful to better understand the advantages of the system

proposed in this work. We have added the comparison below to the Supplementary materials which include the references proposed by referees 1 and 2.

	VLSI 2024 experiment [12]	ISSCC 2024 experiment [14]	Sci. Rep. 2024 experiment [15]	ISSCC 2023 experiment [13]	VLSI 2025 experiment [11]	This work simulation [10]
Technology	65nm CMOS	65nm CMOS	65nm CMOS	65nm CMOS	55nm CMOS	28nm CMOS
Chip area (mm^2)	0.27	1.115	1.8	0.93	0.544	–
Max. order k_{\max}	3	3	2	128	64	–
Max. variables	20	50	20	128	64	–
Max. clauses	91	218	91	1024	256	–
Supported problem types						
CNF	yes	yes	no	yes	yes	yes
XNF	no	no	no	no	no	yes
Solution time						
20/50 var. (us)	7/NA	7/19	$16 \cdot 10^3$ /NA	70/713*	5/45	1/10
Solution energy						
20/50 var. (nJ)	11/NA	8/21	$15 \cdot 10^3$ /NA	105/1098*	59/518	5/30

Supplementary table 3: Comparison of different SAT hardware accelerators when solving random 3SAT problems with 20 and 50 variables. For the RRAM architecture simulations, the semi-analytical energy model does not place hard constraints on the number of variables or clauses. Chip area, k_{\max} and maximum variables and clauses are thus not reported. (*): Solution time and energy for [13] only reported for 60 variables.

As we do not have access to the actual hardware, we are considering the common instances reported in literature, which are random 3-SAT problems with 20 and 50 variables. For these instances, we compare the performance of other accelerator systems to an SRAM-based experimental realization of our in-memory computing architecture reported in [11], as well as simulations of our RRAM architecture, which were previously reported in [10]. For the simulations of our RRAM architecture, we find that it is outperforming other accelerator systems. We want to stress that the advantages of the RRAM system partly arise due to the fact that we compare experimental systems to computer simulations, which can be more optimistic due to the lack of leakage power and other parasitic effects. Moreover, our architecture is using a different memory technology (RRAM) and a smaller technology node (28nm), both of which can result in faster operating speeds and lower energy consumption. For that reason, we believe that our experimental system reported in [8] is a fairer base of comparison. Here, we have reported either on-par or considerably better than other state-of-the-art systems.

While we cannot test our benchmark problems on any of these hardware systems, we want to point out that all systems, except for Snap-SAT and our own architecture, can only support up to 3 literals per clause. This is insufficient to implement the considerably denser pre-processed SAT instances with up to 7 literals per clause. These accelerator systems would thus only be able to implement the sparser un-pre-processed instances and thereby suffer from the same performance penalties that we have observed in our detailed intrinsic comparison (around 30x more iterations and energy to find a solution). In addition, as none of the state-of-the-art systems can support XNF clauses, we expect that

the advantages we observe from translating problems from CNF to XNF in our intrinsic comparison (around 4x fewer iterations and 11x less energy) would also exist compared to the SOTA systems in supplementary table 3.

Considering the data in the table above and the results of our intrinsic comparison, we are therefore confident that our proposed XNF accelerator architecture can attain a performance advantage over existing IMC accelerators reported in the literature.

In response, we have modified parts of the discussion section:

“For SAT problems that can be natively expressed as hybrid XOR--CNF problems, we find that this can reduce the chip area and energy consumption, while also improving the computation speed compared to mapping them to a pure CNF representation. This presents an advantage over existing SAT hardware accelerators, which can only solve problems in pure CNF formulation. When tackling pure CNF problems, the IMC architecture in FIG. 2 has previously demonstrated that it can outperform comparable SAT accelerators (see Supplementary materials). As shown in our intrinsic comparison in FIG. 4, the ability to implement XOR clauses can provide an additional order-of-magnitude improvement in computation speed and energy efficiency.

Moreover, the crossbar array embedding depicted in FIG. 2 can, in principle, support dense XOR and CNF clauses with as many literals as there are variables. Our experimental proof of concept successfully demonstrates this for a hybrid XOR--CNF problem with up to five literals per clause, which can be extended to even more complex clauses. This allows our architecture to additionally leverage the advantages of SAT pre-processing techniques, which tend to trade increased algorithmic efficiency with a higher density of literals per clause (see TABLE 2 in Section 4.2). By combining these advantages, we find that our proposed accelerator can outperform state-of-the-art SAT solvers running on digital computers in terms of computation speed and energy consumption.”

We have also added the above comparison to Section 4 in the Supplementary materials.

* What is the improvement of the proposed WalkSAT-XNF algorithm over existing CNF and XNF solvers if both implemented on CPUs?

We thank the referee for the question. We would like to highlight that the key measures of improvement are the median TTS and ETS metrics reported in TABLE 1 in the manuscript, where the TTS used for WalkSAT-XNF is based on the hardware-level estimate, which is the product of ITS and time per iteration, where we assume a latency of 6 nanoseconds for our

hardware. Here, we present the TTS estimated from the CPU runtime of WalkSAT-XNF; the ETS estimates are shown in FIG.6 in the manuscript.

The average ITS and TTS comparisons for the instances in the XNF-PP class, based on measurable metrics, are as follows.

XNF-PP ITS average	WalkSAT-XNF	xfnSAT	ITS improvement (xfnSAT/ WalkSAT-XNF)
McEliece	77,304	5,627,260,452	72,794
MDP	51,730,543	164,732,553	3
AES	22,943,569	522,249,283	23

XNF-PP TTS average	WalkSAT-XNF (IMC)	WalkSAT-XNF (CPU)	xfnSAT (CPU)	CryptoMiniSat (CPU)
McEliece	0.0005	0.034	1354.026	0.005
MDP	0.3104	84.983	13.442	0.008
AES	0.1377	26.193	66.867	0.017

The ITS and TTS comparisons among the CNF solvers are as follows.

CNF-PP ITS average	WalkSAT-XNF	WalkSAT-SKC	ITS improvement (WalkSAT- SKC/ WalkSAT-XNF)
McEliece	292,611	519,921	1.8
MDP	47,131,444,460	735,008,980,813	15.6
AES	36,293,586	808,645,266	22.3

CNF-PP TTS average	WalkSAT-XNF (IMC)	WalkSAT-XNF (CPU)	WalkSAT-SKC (CPU)	Kissat (CPU)
McEliece	0.0018	0.148	0.074	0.008
MDP	282.7887	352,646.54	382,688.82	0.038
AES	0.2178	28.1	165.898	0.027

As shown in the tables above, the TTS performance of the CPU-based WalkSAT-XNF lies between that of SLS solvers and CDCL solvers, with CDCL-based solvers (CryptoMiniSat, Kissat) often demonstrating better performance. We would like to highlight that our key argument is not based on TTS metric based on CPU implementation, but more importantly, on energy efficiency, as measured by ITS combined with the energy consumed per iteration, that is reported in Fig.6 and TABLE 1 in the manuscript.

5. The evaluation of the detailed improvement of in-memory architecture is also missing, which makes the necessity of using in-memory computing unclear. Please provide the

improvement of the proposed architecture compared with implementing WalkSAT-XNF on CPUs.

-> We thank the referee for raising this point. The table below reports the average TTS grouped by instance class and formulation class, along with their improvements measured on the CNF and XNF representations. The TTS metric (TTS(CPU)) is measured from a CPU-based implementation written in the Rust programming language. The TTS metric (TTS(IMC)) is measured by iterations-to-solution multiplied by a latency of 6 nanoseconds. As reported in the Speedup columns, TTS measured by IMC hardware demonstrates orders-of-magnitude improvement. We believe that reporting the following would help strengthen the necessity of introducing IMC hardware.

Instance class	CNF			XNF		
	TTS (CPU)	TTS (IMC)	Speedup	TTS (CPU)	TTS (IMC)	Speedup
McElice	0.15	0.0018	84	0.03	0.0005	73
MDP	352,646.54	282.7887	1247	84.98	0.3104	274
AES	28.10	0.2178	129	26.19	0.1377	190

The table above is included as Table 2 in the Supplementary Material.

6. What is the maximum problem size (in terms of variables and clauses) that the proposed RRAM crossbar arrays can support considering device non-idealities?

We thank the referee for raising an important point and agree that it would help to be more specific in the manuscript about the dimensions and scalability of RRAM crossbar arrays. Individual RRAM arrays demonstrated in the literature [e.g., An analog-AI chip for energy-efficient speech recognition and transcription | Nature] can potentially support up to 256 variables and 512 clauses. However, crossbar arrays can also support problems larger than this by utilizing tiling architectures, where problems are split across different arrays. State-of-the-art systems have demonstrated this for large-scale problems using up to 140 million RRAM memory cells. We have also recently proposed an architecture that can exploit the sparsity of SAT problems to map problems with up to 1000 variables onto a single chip with an area of $\sim 50\text{mm}^2$.

In order to address this comment, we have changed parts of our previous Discussion section, where the scalability of SAT hardware accelerators is discussed:

“Another challenge relates to the scalability of the IMC hardware. Crossbar arrays are realistically limited in size by parasitic effects and signal drop-off to a few hundred rows and columns. Using our architecture, current IMC hardware could support SAT problems with up to ~ 250 variables and ~ 500 clauses within a single array.”

7. What is the solvability performance of the proposed algorithm and architecture?

-> We thank the referee for the question. The architecture maintains a 1-to-1 correspondence with the proposed algorithm, and the average solvability of the proposed algorithm (which therefore also reflects the architecture) is summarized in Supplementary table S.1; solvability for all benchmark instances containing both CNF and XOR is 100%, i.e., all problems were solved at every repeat.

To evaluate hardware-based solvability, we tested three representative instances, as shown in the table below. The instance par8-1 (XNF-PP), detailed in Section 4.2 of the manuscript, is included among them.

	number of variables	number of CNF clauses	number of XOR clauses	number of repeats	max number of iterations
par8-1 (XNF-PP)	13	41	1	500	2000
par8-2 (XNF)	31	30	23	50	3000
par8-3 (XNF)	31	30	23	50	3000

Each instance was solved successfully in every repeated attempt, yielding a solvability rate of 100%; the first instance par8-1 was solved 500 out of 500 runs, and other two instances were solved 50 out of 50 runs. This result indicates reliable solvability performance across both the architecture and the hardware experiment, and we are confident that using in-memory computing hardware for solving XOR+CNF problems is feasible.

Reviewer #2 (Remarks to the Author):

In this paper, the authors introduced a WalkSAT-XNF, a stochastic local search (SLS) heuristic designed to natively handle hybrid XOR–CNF Boolean formulas. The key idea is to extend the classical WalkSAT framework to directly evaluate both CNF and XOR clauses without requiring conversion into pure CNF form. The authors further demonstrate how this algorithm can be efficiently implemented using analog crossbar arrays based on memristive devices, leveraging their parallelism for gradient computation in variable selection. However, my major concern is that the experimental evaluation lacks sufficient

comparison with modern XOR-CNF SAT solvers (aaai.v33i01.33011592)(978-3-030-53288-8_22)(s11786-024-00594-x), especially those that already support native XOR clause handling. The lack of relevant comparisons also worries me about the innovation level of the algorithm. Secondly, there is also a lack of comparison with existing IMC-based solvers (ISSCC42615.2023.10067380) in terms of hardware; only comparing with CPU-based solvers is limited.

-> We appreciate the reviewer's careful assessment and thoughtful feedback. Below, we respond to each comment with clarifications and additional detail.

Below are the suggestions:

1. The paper primarily compares WalkSAT-XNF with traditional CNF-only solvers (e.g., Kissat, WalkSAT-SKC), which is insufficient. A meaningful comparison should include Advanced XOR-SAT solvers (aaai.v33i01.33011592 / 978-3-030-53288-8_22 / s11786-024-00594-x), specifically designed to handle CNF-XOR formulas arising from hashing-based approximate model counting. Without such comparisons, it is unclear whether the observed performance improvements are due to the IMC architecture or simply because the problems are better suited to XOR-enhanced formulations.

-> We thank the referee for the suggestion to compare WalkSAT-XNF against advanced XOR-SAT solvers. We agree that comparing our results within the context of recent work on CNF+XOR solving is important to properly assess the contribution of our approach.

The two referenced papers (aaai.v33i01.33011592 / 978-3-030-53288-8_22) focus on improving the efficiency of hashing-based algorithms for model counting. They build on the BIRD (with the second paper introducing BIRD2) architecture to efficiently inject XOR-constraints during the BLAST step that arises from the cell-forming routine in approximate model counting. The related architecture has now evolved to ApproxMC6 (<https://github.com/meelgroup/approxmc>). They both rely heavily on an XOR-aware CDCL solver, CryptoMiniSat, once the XOR+CNF formulation is reconstructed after in-processing. Our solver, WalkSAT-XNF, which operates in a stochastic local search regime, is able to natively operate on XOR constraints, without requiring reformulating the problems into purely CNF-based instances. We have benchmarked against CryptoMiniSat in our evaluation, as it serves as a strong representative of the XOR-aware CDCL solver class. We believe that comparing WalkSAT-XNF with CryptoMiniSat constitutes a fair

comparison, as it constitutes the backend for the 3 referenced approximate model counting solvers.

The third work referenced (s11786-024-00594-x) explores a DPLL-style algorithm tailored for 2-XNF (XOR-OR-AND normal form), where each clause is a disjunction of at most two literals. This is an interesting theoretical direction that differs from our approach both in scope and solver architecture.

We attempted to translate XOR+CNF formulae into 2- XOR-OR-AND normal form in order to benchmark the instances used in our paper by using the solver introduced in `s11786-024-00594-x`. The translation proceeded as follows: a k-literal CNF clause is reduced to (k-2) 3-literal clauses with k-3 auxiliary variables. Each 3-literal clause is then converted to three 2-literal clauses with the addition of auxiliary variables. However, this transformation does not guarantee satisfiability of the resulting 2-literal clauses, since there is no known satisfiability-preserving reduction. Despite this limitation, we adopt the following conversion of the type $(x \vee y \vee z) \rightarrow (x \vee a) \wedge (y \vee \neg a) \wedge (z \vee \neg a)$, with a being an auxiliary variable introduced. The XOR clauses do not require any conversions and hence are omitted from the plot. We have observed that this conversion is not suitable for certain problem families, such as the McEliece class, which admit a unique solution. We believe that this uniqueness poses a problem in the converted form. For instance, if $(x,y,z)=(0,1,0)$ is the unique solution to the instance, the 2-literal clauses are unsatisfiable. In fact, we observed that all McEliece instances reduced to contain 2-literal CNF clauses become unsatisfiable.

In the reverse direction, we attempted to translate existing 2-XOR-OR-AND normal form instances accessible from the link <https://github.com/j-danner/2-Xornado/tree/main/tests/2xnfs> into XOR+CNF formulae. This collection of instances is translated into a set of CNF+XOR instances suggested by Remark 5.1 in (s11786-024-00594-x), and the resulting statistic is reported in the plot below.

Here, we observe a significant increase in the size of the instances, with the total number of clauses often doubling the number obtained in the XOR+CNF formulation. In our case, the solver is deployed on specialized in-memory computing hardware, where the number of clauses are constrained by the physical capacity of the system. Reducing high-order CNF clauses to quadratic form significantly increases problem size, potentially beyond the hardware capacity. To clarify these limitations, we have added the following explanation in the Discussion section of the revised manuscript:

“Another challenge relates to the scalability of the IMC hardware. Crossbar arrays are realistically limited in size by parasitic effects and signal drop-off to a few hundred rows and columns. Using our architecture, current IMC hardware could support SAT problems with up to ~250 variables and ~500 clauses within a single array.”

Furthermore, our hardware can support clauses with multiple literals, applying this transformation undermines its architectural advantages. To reinforce this point, we have strengthened TABLE 2 in Section 4.2 to show the clause degrees that our hardware can support.

We revised Section 2 of the Supplementary material to include discussions of these related works and clarify the distinctions in methodology and applicability. We also highlighted that the observed performance improvements stem from both the IMC-inspired architecture and the SLS-based approach CNF+XOR solving using the TTS metric, based both on the IMC and CPU implementations (Section 1 of the Supplementary material).

2. The paper claims that WalkSAT-XNF is uniquely suitable for IMC acceleration, but does not clearly explain why other algorithms (e.g., CDCL + Gauss-Jordan elimination) cannot be adapted to similar architectures.

-> We thank the referee for the question; we think this is an important point to clarify. There appears to be a misunderstanding. We do not want to claim that WalkSAT-XNF is uniquely suitable for IMC acceleration. While the design of dedicated CDCL or GJE accelerators would be a large undertaking and thus outside the scope of this work, we agree with the referee that there is a large potential for these algorithms to be accelerated by IMC. In the Discussion section of our manuscript, we had thus previously mentioned CDCL and GJE as promising algorithms that should be investigated in the future. Given the remark by the referee, we feel that it is important to better stress that WalkSAT-XNF is not uniquely suitable for IMC and we see potential in exploiting IMC for other types of algorithms.

In response, we have changed the last paragraph of the Discussion section:

“The WalkSAT-XNF heuristic is an evolution of the CNF-specific WalkSAT heuristic and does not differentiate between XOR and CNF clauses for the purposes of variable selection. Based on the insights from this work, it could be possible to use IMC hardware for accelerating algorithmically efficient heuristics that include more sophisticated clause differentiation (e.g., by pre-solving the XOR clauses using Gauss--Jordan elimination).”

3. No discussion is provided regarding why certain problem classes benefit more from one approach over another, nor is there a breakdown of structural features (e.g., XOR clause density, clause length, variable dependencies) that influence solver performance.

-> We thank the referee for raising the question; we agree that including such discussions may help strengthen the manuscript. We aim to compare our observations with the work ‘The Hard Problems Are Almost Everywhere For Random CNF-XOR Formulas (Dudek et al., IJCAI-17, 2017, pp. 600–606) as a baseline. This work provides empirical findings on the performance of CryptoMiniSat using random XOR+CNF formulae, particularly focusing on the structural properties of XOR clauses. We use instance size, XOR clause degree, and clause-variable interdependency as metrics to gauge solver performance, and analyze the consistency of our findings with those reported by Dudek et al.

First, we observe how instance size impacts solver performance. Regardless of the problem classes, the instance sizes are the most dominant factor of the energy efficiency of our solver. The relationship between energy-per-iteration and instance size is evident in the top row of the plot below. In addition, there is a strong positive relation between instance size and ITS (and therefore ETS), as shown in the bottom row of the figure below.

Secondly, we observe a positive relation between the degree of XOR clauses to ITS, shown in the plot below. Specifically, higher XOR clause degrees tend to increase problem hardness, leading to larger ITS values. This observation aligns with empirical findings by Dudek et al.,. However, it is important to note that this is not an entirely direct comparison, as our benchmarks are based on structured problem instances, whereas the results in the work by Dudek, et al.’s work are based on random problem instances under controlled settings. Additionally, we observe that the XOR clause degrees of the XNF-PP class instances tend to be greater than those of the XNF class instances. The increase in degree with respect to CNF clauses is also observed with preprocessing, yet, the degree of CNF clauses appears to have a weaker correlation with ITS.

Finally, variable dependency--measured by the average frequency of clauses where each variable appears in--is negatively correlated with ITS. Although the correlation signal is weaker than other cases, a generally decreasing trend is still observed. We believe the

trend shown in the plot below does not significantly violate the findings made in the aforementioned work by Dudek, et al. particularly when the XOR variable-probability is less than 0.2. In this region, the hardness and variable-XOR interdependency exhibits a concave pattern.

We have revised the manuscript to include the observations related to the instance size and clause degrees in Section 1 of the Supplementary material. The discussion regarding the variable-clause interdependency is not included, since the variable-XOR clause interdependency in our benchmark set does not span beyond 0.2, rendering the comparison inconclusive.

4. The paper emphasizes the acceleration and energy efficiency advantages of the IMC architecture over traditional CPUs, but does not compare with other IMC-based SAT accelerators such as Snap-SAT (ISSCC42615.2023.10067380).

-> We agree with the referee that a comparison to other SAT hardware accelerators is needed to better highlight the benefits of our proposed accelerator architecture. We would like to point the referee to our answer to a similar question by referee #1, where we describe the results of a comparison of our system against other state-of-the-art SAT hardware accelerators. We have also included Snap-SAT in the comparison, as suggested by the referee.

5. Figure 2 of the paper illustrates the hardware module partitioning of WalkSAT-XNF, including clause evaluation, make/break computation, WTA, and other components, but provides no detailed description of its control logic or synchronization mechanisms. How do these modules coordinate with each other? Is there any control bottleneck?

->We agree with the referee that a more detailed description of the operation of our architecture would be helpful, e.g., in identifying potential bottlenecks. We have added additional details to the Methods section to better describe the dataflow and operation during a single iteration. The entire system is driven by a central clock signal, where the signal is distributed to the different circuit elements by a clock tree to achieve synchronization. A full iteration takes three clock cycles in total. During the first clock cycle, the signals are applied to the first crossbar array and the output signals are analyzed using the readout circuit. During the second clock cycle, the second crossbar array is operated in the same way. During the third clock cycle, the WTA operation is performed, and the register state is updated. Once the register has been set with an initial variable configuration, the circuit will continuously repeat this flow, until the register is read out again. With this continuous autonomous operation, there is no need for complex control logic. We cannot identify a significant control bottleneck.

In response, we have extended the Methods section, where we describe the modeling of the hardware:

“The circuit is driven and synchronized by a central clock signal, where the signal sequentially progresses through the individual circuit blocks shown in Fig.2. A single iteration of WalkSAT-XNF is performed in three clock cycles. During the first clock cycle, the signals are applied to the first crossbar array and the output signals are analyzed using the readout circuit. During the second clock cycle, the second crossbar array is operated in the same way. During the third clock cycle, the WTA operation is performed, and the register state is updated. The combined latency of these components per iteration of WalkSAT-XNF was modelled as taking $t_{\text{iter}} = 6 \text{ ns}$. Once the register is initialized, the entire circuit will continuously repeat this flow until a predefined number of iterations is reached or until a satisfying solution has been identified.”

6. WalkSAT-XNF utilizes analog signals to perform parallel evaluation of make/break values, but does not quantify the precision loss of analog components such as ADCs and comparators. What is the impact of different ADC resolutions on ITS (Iterations-to-Solution)?

->We thank the referee for raising an important point. In general, our hardware simulations discretize the analog output signal coming from the crossbar array to mimic the operation of an ADC or a comparator as shown in Fig.3 (i.e., each ADC level is associated with a specific number of true literals computed by the crossbar array). The simulations therefore already include the effects of signal discretization. Within all simulations, we have chosen

an ADC resolution of 4 bits, which corresponds to 16 discrete levels. Given that the maximal number of literals per XOR clause for our benchmark problems is 15, this resolution is sufficiently high to evaluate XOR clauses even under the worst-case scenario, i.e., when all literals are true. We therefore do not expect that the current ADC resolution will have an adverse effect on the performance. That said, we see optimization potential in reducing the ADC resolution. We have analyzed the simulated output signals from the first crossbar for XOR clauses for two exemplary MDP problems in the figure below. We find that the average number of true literals for XOR clauses is considerably below the maximum number of literals. The distribution also shows that it is unlikely to encounter the case where all literals are true. This would allow us to reduce the resolution of the ADC, which would improve the overall energy efficiency of the accelerator. We have tested the effect of reducing the bit resolution on the cumulative success rate by reducing the number of ADC levels. Depending on the instance, we find that the success rate is not significantly affected by as few as 4-5 ADC levels. For a lower number of ADC levels, we see that the success rate will diminish. Notably, for par8-2, we see that even a 2-level ADC (effectively a comparator) is still able to solve the example problem, albeit in a considerable number of trials.

In response, we have added an additional remark to the Discussion section:

“As we show in the Supplementary materials, additional energy savings can also be achieved by reducing the resolution of the ADC.”

We have also added a remark to the Methods section, where we explain the choice of the bit resolution for the ADCs:

“Based on the maximum number of literals for the benchmarks problems (see Table 2) we assume an ADC bit resolution of 4 bits, which can support clauses with up to 15 literals.”

Finally, we have added a new section (Section 4) to the Supplementary materials, where we discuss the data shown above.

REVIEWER COMMENTS

Reviewer #1 (Remarks to the Author):

“The authors have addressed most of my questions, but a few concerns remain not well resolved.”

“The maximum problem size that the proposed analog CiM array can support is limited primarily by the variations and on/off ratios of RRAMs as well as the non-idealities of the ADCs, instead of by the RRAM array size. Therefore, the claim of supporting up to 250 variables and 500 clauses is not convincingly substantiated.”

We agree with the reviewer that we should clarify the feasibility of using large-scale in-memory computing systems in our architecture. As the reviewer correctly remarks, scaling of our architecture requires that we can perform large-scale matrix-vector multiplications (MVMs) using analog compute in memory (CiM) arrays. CiM-based SAT accelerators, such as the ones in [11, 14], have demonstrated successful operation using arrays as large as 128 by 1024. Sensitivity studies of device non-idealities in RRAM-based CiM hardware have shown the feasibility of solving SAT problems with up to 200 variables and 860 clauses [36]. CiM arrays based on emerging memories (such as RRAM or phase change materials) for performing MVM operations have also been reported in the literature for array sizes of up to 2048 by 512 [46] for AI inference, where the layer-to-layer connections are represented by dense matrices, i.e., a very high on/off ratio for the memory cells. Such systems achieve large-scale dense matrix-vector operations, despite programming variations of individual memory cells or non-idealities in the sensing circuits, e.g., in the ADCs. It must be noted that, compared to AI inference tasks, the requirements of our architecture for the CiM array are considerably more relaxed. Contrary to the dense MVM operations in AI workloads, our architecture maps SAT problems to matrices that are relatively sparse. As shown in Table 2, the average on/off ratio for each individual array row does not exceed 20%. Moreover, compared to AI workloads, our architecture does not require multi-bit precision for the input data or the memory cells. The requirements on memory cell variations and the dynamic range of the sensing circuits are thus considerably more lenient. We are therefore confident that large-scale state-of-the-art CiM systems, such as the ones reported in [46], can also be used in our architecture. This is also supported by a sensitivity analysis that we have added to the Supplementary Information.

In response to the reviewer’s comment, we have modified the discussions section to incorporate the reviewer’s remarks related to non-idealities and dense matrices:

“One challenge in realizing performance enhancements for industrial applications pertains to the scalability of IMC hardware. Crossbar arrays are limited in size, for example, by parasitic effects, signal drop-off, and non-idealities, to a few hundred rows and columns. Current IMC hardware capable of dense matrix--vector operations could support the computations in our architecture for SAT problems with up to ~250 variables and ~500 clauses within a single array [46].”

We have also added additional details to section 2.4, where we discuss the dominant types of non-idealities in CiM hardware:”

“As with other mixed-signal computing systems, realizing WalkSAT-XNF in hardware requires it to be sufficiently resilient against hardware non-idealities in the analog circuits. Studies have identified variations in the RRAM cells and noise in the crossbar array's analog readout circuit as the dominant non-idealities that can result in a deterioration in performance [36].”

We have also added a new section 5 to the Supplementary Information, where we discuss and evaluate the sensitivity of our proposed architecture to hardware non-idealities, including RRAM variations and ADC noise.

“The comparisons with the state-of-the-art CiM-based SAT accelerators are not apple-to-apple. In Supplementary Table 3, as the authors have mentioned, the proposed work uses a more advanced technology node, and thus the results are not directly comparable. The source of the improvement on pure CNF problems against other baselines is also unclear. Besides, the authors claimed “our experimental system reported in [8] is a fairer base of comparison”, yet [8] is not compared. The use of [10] to cite the proposed work within the table is also very confusing.”

We thank the reviewer for raising this point. Clearly, we have not done a good enough job in explaining the comparison in Tab.S3, and we agree that we should clarify this further. As correctly noted by the reviewer, the SAT accelerators cited in Tab.S3 are experimental proof-of-concept systems, whereas our simulations aim to predict performance attainable by a hardware accelerator. Compared to our simulations, experimental proof-of-concept systems sacrifice some of their performance, e.g., by using larger technology nodes and potentially less energy-efficient but more readily available memory technology. Moreover, these chips are typically operated in testing environments, which can produce additional power losses, e.g., due to test circuit boards or external signal sources. In the comparison in Tab.S3, we have therefore included both simulations (based on the work reported in [8]) and a proof-of-concept system [14] for a CNF-only SAT solver using our IMC approach.

Comparing [8] and [14] shows that an optimized CiM SAT accelerator can potentially improve compute speed and energy efficiency by a factor of 5x and 10x, respectively, over the proof-of-concept system. When comparing [14] to the other hardware systems, we can also gain insights into how our CiM approach compares to other state-of-the-art systems. [12] is a system that is very similar in capability to [14], as it can solve k-SAT problems with arbitrary numbers of literals per clause using a WalkSAT-like heuristic. We find that our approach is ~15x faster and ~2x more energy efficient. Compared to [10] and [11], we find that [14] is up to 2.4x slower and up to 25x less energy efficient for 3SAT problems. It is important to note that [10,11] have been specifically optimized for solving 3SAT problems, i.e., problems with only 3 literals per clause. Because of this, whereas [12,14] could solve our benchmark problems using the more efficient pre-processed representation ('CNF-PP' in Fig.S1), [12] and [14] would have to use the non-pre-processed problem representation ('CNF' in Fig.S1). As our data in Fig.S1 shows, these problems are larger in size and more complex to solve with SLS solvers, which results in a ~30x increase in computation time and energy consumption. For our benchmark problems, the benefits of the more energy efficient 3SAT solvers in [10,11] would thus be offset by the overhead of mapping to 3SAT. From the comparison in Tab.S3, we therefore estimate that our CiM approach is either better, or on-par compared to other state-of-the-art SAT accelerators for our benchmark problems in CNF.

Given the reviewer's comment, we have decided to rewrite large parts of section 3 in the supplementary materials using our explanation above.

Reviewer #2 (Remarks to the Author):

"The authors have provided detailed responses to the initial reviews and have made substantial revisions to the manuscript. However, several points still require clarification or further analysis to strengthen the paper:"

"1. Since CDCL+GJE could also be accelerated via IMC, please elaborate on why WalkSAT-XNF was chosen as the first target instead of other algorithms. Give a brief discussion of the relative feasibility and potential performance of IMC-accelerated CDCL+GJE."

We thank the reviewer for the suggestion and gladly provide further clarification.

WalkSAT-XNF was designed because its WalkSAT-type algorithm provides a natural embedding of the problem in a matrix representation, which can be directly utilized for

gradient computation and for evaluating clauses with IMC hardware. Moreover, WalkSAT-XNF's relatively low computational complexity enables us to translate all its computational steps directly into analog circuits, which can perform all the computations during an iteration within just three clock cycles. The continuous and autonomous operation of this circuit avoids the need for complex control circuits, and the time and energy overhead of frequent memory access or communication with a co-processor, resulting in fast and efficient computation. Moreover, the entire problem can be embedded into the crossbar array without requiring frequent reprogramming of the memory cells. Given the reviewer's comment, we have added the following to section 2.3 to better explain our choice:

“Crucially, the relative simplicity of WalkSAT-XNF enables us to map every computational step to an equivalent analog circuit, enabling rapid continuous computation. As with other IMC concepts [8,34], the crossbar arrays in Fig.2 enable parallel gradient computations for both the CNF and XOR clauses within a single clock cycle. Performing an entire operation of WalkSAT-XNF is achieved within just three clock cycles, without the need for a complex control system, while also circumventing frequent time-intensive communication with external co-processors or memory systems. Both XOR and CNF clauses can be evaluated using the same array, allowing for an area-efficient design. Moreover, the crossbar array can implement a number of literals per clause that is equal to the number of variables, hence supporting highly complex clauses common in industry workloads.”

Regarding the comparison to CDCL+GJE, it appears as if our previous reply has created the impression that it is easy to accelerate these methods with IMC hardware. We want to clarify that there is currently no architecture reported in the literature that can implement CDCL solvers with native XNF support using IMC hardware. We believe that this is due to CDCL algorithms being more complex compared to WalkSAT-like algorithms, which creates some yet unsolved design challenges. For example, it is unclear whether the computational steps can be mapped to a continuously operating analog circuit that could provide rapid computation as with WalkSAT-XNF. Moreover, it is unclear whether such a CDCL IMC accelerator would avoid frequent time-intensive communication with co-processors and memory systems and whether it would circumvent the need for frequent reprogramming of the IMC hardware. A performance comparison with WalkSAT-XNF would thus require us to first conceive, design, model, and benchmark an architecture, which would be similar in scope to the design of WalkSAT-XNF. If it is possible to design such an accelerator, we think that this would be deserving of a separate publication, given the novelty and considerable design challenges.

We would also like to clarify that we did not want to imply that Gauss-Jordan elimination (GJE) is easily implementable on IMC hardware. GJE requires altering many components

within the matrix due to the elementary row operations, such as row swapping, adding/subtracting a subset of rows to replace a row within the matrix. Implementing this on an IMC crossbar array would necessitate multiple reprogramming steps--even if polynomially bounded--which is not ideal. Hence, for the CDCL combined with GJE--as implemented in CryptoMiniSAT--we believe that GJE should be performed independently on systems that can alter computational components efficiently and perform complex logic, which will require data transfer between the accelerator and a co-processor.

Considering the origin of GJE, as discussed in "Extending SAT Solvers to Cryptographic Problems. SAT 2009: 244-257 by Mate Soos, Karsten Nohl, Claude Castelluccia", it is primarily used for unit propagation after constructing a system of linear equations derived from XOR clauses. By taking an assumption set (i.e., assuming a subset of variable assignments within a search tree), unit propagation can be attempted whenever a single equality (i.e., an XOR clause) contains exactly one unassigned variable. Meanwhile, ongoing work for CDCL algorithms using hardware accelerators for CNF clauses--such as "SAT-Accel: A modern SAT solver on an FPGA. In Proceedings of the 2025 ACM/SIGDA International Symposium on Field-Programmable Gate Arrays (FPGA '25), 234-246. ACM by Lo, M., Chang, M.-C. F., & Cong, J."--has demonstrated novel techniques for accelerating propagations, which is referred to as the main bottleneck in conventional CDCL solvers. In their approach, each variable is assigned a unique bit-wise identifier derived from binary representation of an integer. Each clause is compactly represented by a signature, computed as the bitwise XOR of all variable IDs in that clause, along with a counter tracking the number of unassigned variables. Whenever a variable is assigned (i.e., joins the assumption set), its identifier is XORed with the current clause signature, and the counter is decremented. When only one variable remains unassigned, the resulting signature directly identifies that variable, enabling efficient unit propagation. This method relies on fixed-width XOR and simple counter operations. Although representing variable indices as bit-wise identifiers introduces a logarithmic overhead of $\log(n)$ bits the overall scheme appears suitable for hardware implementation. Moreover, both forward propagation and backtracking introduced in this method can reuse the same clause signatures (i.e., allowing to reuse the same clause signature to perform both operations), further simplifying the logic. Given that our IMC hardware natively supports XOR operations, we believe that our IMC hardware could be used to accelerate part of the computational approach presented by Lo et al. That said, as explained above, an evaluation of the feasibility and performance of this concept would require the design of an entirely new accelerator architecture, which is beyond the scope of our current manuscript.

In response to the reviewer's comment, we have added reference [51] to the work by Lo et al. mentioned in the explanation above.

“2. The comparison table shows that other accelerators do not natively support XNF. However, is it not possible to pre-process the XNF problems into a low-literal CNF formulation to run on these accelerators? If so, what would be the expected performance penalty, and how would it affect the overall advantage claimed for your architecture?”

We thank the reviewer for raising this very important point. As the reviewer correctly points out, our benchmark problems can be represented both as XNF problems and as CNF problems. Moreover, the problems can be represented as low-degree and high-degree problems after applying pre-processing techniques. In our benchmarks, these 4 different problem representations are named ‘CNF’, ‘CNF-PP’, ‘XNF’, and ‘XNF-PP’, where ‘-PP’ variants tend to involve higher degree counts, as shown in Table 2 in the manuscript. We have compared the performance when solving these different problem representations in Fig.S1 and in Tab.S1. We find that mapping to low-degree CNF problems increases iterations-to-solutions by a factor of ~100 compared to XNF-PP, while also increasing the number of variables and clauses by a factor of ~5. When solving the low-literal CNF problems with the SAT accelerators in Tab.S3, which are using local search heuristic similar to our work, this would result in considerably slower computation times compared to an XNF-native hardware accelerator. Moreover, we want to stress that due to the increase in the number of variables and clauses (see also Tab.2), the size of many of the low-degree SAT problems often becomes too large to fit into current SAT accelerators. Given the reviewer’s comment, we believe it is important to better emphasize these points in our manuscript. In section 3 of the Supplementary Information, we now specifically mention the scenario described by the reviewer:

“Notably, while none of these accelerator devices support XOR clauses, it is still possible for them to solve hybrid XOR--CNF problems by mapping them to a CNF representation.”

We also added more details to explain the performance penalty of mapping to low-degree CNF problems and how that affects the overall performance of the SAT accelerators in Tab.S3.

“It is important to note that these accelerator [12,14] have been specifically optimized for solving 3-SAT problems, that is, problems with only 3 literals per clause. Because of this, whereas the accelerator in Ref. [13] could solve the benchmarking problems in Fig.S1 using the preprocessed representation (i.e., ‘`CNF-PP’ in Fig.S1), the accelerators in Refs. [12,14] would have to use the un-preprocessed problem representation (i.e., ‘`CNF’ in Fig.S1). As our data in Fig.S1 shows, these problems are larger in size and more complex to solve with SLS solvers, which results in an ~30X increase in computation time and energy consumption. The benefits of the more energy-efficient 3-SAT solvers in Refs. [12,14] would

thus likely be offset by the overhead of mapping to a 3-SAT formulation. Based on our comparison, we estimate that our accelerator is either better than, or on-par with, other state-of-the-art accelerators when solving our benchmarking problems in CNF representation. Projecting the performance of solving the benchmarking problems in XNF representation from Fig.S1, an XNF-capable SAT accelerator would attain an additional improvement by an order of magnitude in both runtime and energy consumption over pure CNF SAT accelerators due to the decrease in problem size and the higher algorithmic efficiency. We also note that the number of clauses for our benchmarking instances in CNF representation can be prohibitively large to implement with many of the CNF SAT accelerators listed in Table S3, whereas the XNF problems are sufficiently small to be implemented with existing crossbar arrays.”

“3. For larger crossbar arrays, could the clock tree distribution and signal skew become a challenge for synchronization? Has this been analyzed or simulated?”

The reviewer raises a valid point. The distribution of clock signals can be a challenge and requires careful engineering by circuit designers. In our energy model, we consider energy consumption by clock trees in individual building blocks. We have not simulated and verified the clock distribution for the entire system. That said, there are various hardware systems that demonstrate the feasibility of achieving clock synchronization in accelerator devices using crossbar arrays, including accelerators for SAT problems [11, 14] and AI workloads [45]. Some of these chips, e.g., in [45], achieve synchronous operation across multiple parallel crossbar arrays. Given that, we are confident that clock signal distribution is also feasible within our proposed architecture.

“4. The simulation results are promising, but the reliance on idealistic models (28nm, no leakage) may overstate the advantages.”

We thank the reviewer for raising this point. Clearly, we haven’t done a good enough job in explaining the sources of the differences in performance in Table S3. In particular, our use of the term “leakage” appears to have caused a few misunderstandings about our energy model based on the 28nm TSMC foundry process. We’d like to clarify that our results are based on a commercial foundry process design kit (PDK). This PDK contains detailed models for leakage power in all building blocks, e.g., subthreshold leakage in transistors, which are also accounted for in our energy model. Our simulations are therefore a prediction of a chip fabricated in that foundry technology node. Within the discussion of Tab.S3, “leakage” was used to refer to the fact that many of the chips are proof-of-concept systems. Such systems are operated in testing environments, which can produce an additional energy overhead,

e.g., due to additional losses from test PCBs, cables or external voltage/signal sources. The energy reported for the experimental systems in Tab.S3 can include these overhead losses, which would not be present in a fully integrated system. In contrast, our simulations aim to predict performance attainable by a fully integrated accelerator device and not that of a proof-of-concept system. Within our manuscript, we think it is important to explain this distinction to facilitate a fair comparison. Given the reviewer’s comment, we have decided to rewrite section 3 of the supplementary materials to better explain this point. We now explain the differences between experimental systems and our simulations in table S3 in the following way:

“We note that the SAT accelerators reported in the literature [11-15] are experimental proof-of-concept systems, whereas the goal of our work is to project the performance attainable by a fully integrated accelerator device. Compared to our simulations, these proof-of-concept systems employ larger technology nodes and potentially less-efficient memory technology. Moreover, proof-of-concept systems are typically operated in test environments that can produce additional energy overhead, for example, due to energy losses incurred from test circuit boards, cables, or external signal sources. We have therefore included both simulations using our energy model (based on the work reported in Ref. [10]) and a proof-of-concept system of the SAT accelerator reported in Ref. [11] that uses our IMC architecture to solve CNF problems.”

“Please provide a sensitivity analysis of key non-idealities (e.g., conductance variations, ADC noise) on the solver’s success rate and energy efficiency.”

We thank the reviewer for the suggestion. We have added a sensitivity analysis to section 5 of the Supplementary Information. In this analysis, we investigate the influence of readout noise (e.g., noise from the TIAs, comparators, ADCs), as well as conductance variations of the RRAM cells (i.e., programming variations and thermal/telegraph noise) on the iterations-to-solution (ITS). As established in [18,36], we model the variations in the crossbar array’s bit line (BL) output voltage during each cycle t_k as:

$$V_{BL,i}(t_k) \propto \sum_j \left(G_{ij} + \mathcal{N}(0, \sigma_{d2d,RRAM}) + \mathcal{N}(0, \sigma_{c2c,RRAM})(t_k) \right) x_j + \mathcal{N}(0, \sigma_{c2c,readout})(t_k)$$

Here, G_{ij} are the target conductance values of the RRAM cells and x_j are Boolean input variables. Variations are approximated by Gaussian normal distributions $\mathcal{N}(\eta, \sigma)$ with zero mean and standard deviations $\sigma_{d2d,RRAM}$ (device-to-device RRAM conductance variations during programming), $\sigma_{c2c,RRAM}$ (cycle-to-cycle RRAM conductance variations due to thermal/telegraph noise) and $\sigma_{c2c,readout}$ (cycle-to-cycle variations due to noise in the readout circuit, e.g., TIAs, comparators, ADCs). Cycle-to-cycle variations change every

cycle t_k , whereas device-to-device variations are set initially and remain constant over time. The conductance-dependent RRAM variations $\sigma_{d2d,RRAM}$ and $\sigma_{c2c,RRAM}$ are based on the model in Ref. [18], which is derived from device characterizations of TaO memristors in Ref. [19]. Here, we assume that the conductance of the low resistance state (LRS) is at 2uS and that the high resistance state (HRS) is at 20nS. In our analysis, we then sweep the magnitude of the readout noise $\sigma_{c2c,readout}$ to investigate how much noise in the peripheral readout circuits can be tolerated before performance of WalkSAT-XNF degrades.

In figure (a) below, we show the cumulative success rate for the MDP benchmark instance ‘par8-3’ for different readout noise levels $\sigma_{c2c,readout}$ and find that increasing the noise will eventually lead to a lower success rate. When analyzing ITS of three exemplary MDP instances as a function of the readout noise in figure (b), we find that ITS remains close to the ideal simulations (without RRAM variations and readout noise) until a noise standard deviation of $\sigma_{c2c,readout} \sim 0.15$ (i.e., fluctuations of up to 15% relative to the voltage steps in the ADC). For larger noise, ITS will begin to deteriorate. We have also performed the same analysis for an LRS conductance of 4uS (figure (c)) and 6uS (figure (d)). When increasing the conductance, the relative amount of noise from the RRAM cells decreases, thereby enhancing the signal-to-noise ratio. We observe increased robustness to readout noise of up to $\sigma_{c2c,readout} = 0.25$, which also increases energy per iteration by up to 10 percent due to the increase in current flowing through the RRAM cells. We want to stress that readout cycle-to-cycle fluctuations of over 10 percent are larger than what we observe in our IMC hardware. Based on circuit simulations, we expect fluctuations in the range of a few percent at most.

This is also in agreement with our data in Fig.3, where we wanted to experimentally verify whether variability in the IMC hardware significantly affects the performance of WalkSAT-XNF. In our experiments, we performed the analog computations within WalkSAT-XNF

(clause evaluation and make/break value computation) on an actual RRAM chip. The RRAM cells in our test chip were programmed with a relatively large variability of $\sim 10\%$. Our results show that, despite the RRAM variations and despite readout noise present in our test chip, we can evaluate clauses with just $\sim 1\%$ error. When using the hardware to solve XNF problems, the success rate and computation time agree well with our ideal noiseless simulations, showing that errors due to device non-idealities do not accumulate significantly enough to affect the solver’s performance. To further verify this, we show additional experiments on another IMC chip for the ‘par8-2’ and ‘par8-3’ benchmark problems below. We can again see good agreement between ideal noiseless simulations and the hardware experiments. Our sensitivity analysis and our hardware experiments thus indicate that WalkSAT-XNF is sufficiently resilient to hardware non-idealities. We believe this to be due to the binary RRAM weights and the binary input states. The results of the crossbar array operation are thus integer values, whose discrete nature is significantly more robust to noise compared to continuous floating-point operations.

In response to the reviewer’s remarks, we have added a new section 5 to the Supplementary Information based on our reply above. We have also added additional details to section 2.4, where we discuss the dominant types of non-idealities in CiM hardware:

“As with other mixed-signal computing systems, realizing WalkSAT-XNF in hardware requires it to be sufficiently resilient against hardware non-idealities in the analog circuits. Studies have identified variations in the RRAM cells and noise in the crossbar array’s analog readout circuit as the dominant non-idealities that can result in a deterioration in performance [36].”

“This observation is also supported by a simulation-based sensitivity study, the results of which are presented in the Supplementary Information. We believe this robustness to be due to the fact that the weights and the input states in our architecture are binary. The results of

the crossbar array's operations are discrete integer values, thereby providing additional robustness against noise, compared to, for example, floating point operations.”

References (as numbered in the revised manuscript and the Supplementary Information):

[8] G. Pedretti, F. Böhm, T. Bhattacharya, A. Heittman, X. Zhang, M. Hizzani, G. Hutchinson, D. Kwon, J. Moon, E. Valiante, I. Rozada, C. E. Graves, J. Ignowski, M. Mohseni, J. P. Strachan, D. Strukov, R. Beausoleil, and T. V. Vaerenbergh, Solving Boolean satisfiability problems with resistive content addressable memories, *npj Unconventional Computing* 2, 7 (2025).

[10] Q. Zhang, S. Su, Z. Liu, H.-C. Cheng, Z. Qiu, M. Palaria, J. Ye, D. Meng, B. Chen, S. Hossain, W. Wu, and M. S.-W. Chen, A stochastic analog sat solver in 65nm CMOS achieving 6.6 μ s average solution time with 100% solvability for hard 3-sat problems, in 2024 IEEE Symposium on VLSI Technology and Circuits (VLSI Technology and Circuits) (2024).

[11] C. Shim, J. Bae, and B. Kim, 30.3 VIP-Sat: A Boolean Satisfiability Solver Featuring 5 \times 12 Variable In-Memory Processing Elements with 98% Solvability for 50-Variables 218-Clauses 3-SAT Problems, in 2024 IEEE International Solid-State Circuits Conference (ISSCC) (IEEE, 2024) pp. 486–488.

[12] S. Xie, M. Yang, S. A. Lanham, Y. Wang, M. Wang, S. Oruganti, and J. P. Kulkarni, 29.2 Snap-SAT: A One-Shot Energy-Performance-Aware All-Digital Compute-in-Memory Solver for Large-Scale Hard Boolean Satisfiability Problems, in 2023 IEEE International Solid-State Circuits Conference (ISSCC) (IEEE, 2023) pp. 420–422.

[14] T. Bhattacharya, D. Kwon, G. Hutchinson, X. Zhang, I. Rozada, and D. Strukov, A Fully Integrated Mixed-Signal Compute-In-Memory Accelerator for Solving Arbitrary Order Boolean Satisfiability Problems, in 2024 IEEE Symposium on VLSI Technology and Circuits (VLSI Technology and Circuits) (2025).

[18] L. Zhao, L. Buonanno, A. Natarajan, J. Ignowski, and G. Pedretti, Noise aware finetuning for analog non-linear dot product engine, in *NeurIPS 2024 Workshop Machine Learning with new Compute Paradigms* (2024).

[19] X. Sheng, C. E. Graves, S. Kumar, X. Li, B. Buchanan, L. Zheng, S. Lam, C. Li, and J. P. Strachan, Low-conductance and multilevel CMOS-integrated nanoscale oxide memristors, *Advanced Electronic Materials* 5, 1800876 (2019).

[36] A. Heittmann, M. Hizzani, and J. P. Strachan, in 2025 IEEE International Symposium on Circuits and Systems (ISCAS) (2025) pp. 1–5.580

[46] S. Ambrogio, P. Narayanan, A. Okazaki, A. Fasoli, C. Mackin, K. Hosokawa, A. Nomura, T. Yasuda, A. Chen, A. Friz, M. Ishii, J. Luquin, Y. Kohda, N. Saulnier, K. Brew, S. Choi, I. Ok,

T. Philip, V. Chan, C. Silvestre, I. Ahsan, V. Narayanan, H. Tsai, and G. W. Burr, An analog-AI chip for energy-efficient speech recognition and transcription, *Nature* 620, 768-775 (2023)

[51] M. Lo, M.-C. F. Chang, and J. Cong, SAT-Accel: A modern sat solver on a FPGA, in *Proceedings of the 2025 ACM/SIGDA International Symposium on Field Programmable Gate Arrays, FPGA '25* (Association for Computing Machinery, New York, NY, USA, 2025) p. 234–246.608

Reviewer #1:

The authors have made substantial improvements in the discussion of hardware non-idealities as well as in the comparison with the state-of-the-art works, and have addressed my previous concerns. One remaining issue is the reference numbering. The authors note that the references in the response letter are as numbered in the revised manuscript and the Supplementary Information. However, I found that references [8] and [14] in the response letter correspond to [10] and [11] in the revised Supplementary Information. This inconsistency confuses me. Please make sure that the reference numbering is consistent and correct across the main manuscript and the Supplementary Information.

-> Response:

We thank the reviewer for the positive assessment of our revisions. We sincerely thank the reviewer for the time and thoughtful feedback provided throughout the review process.

We appreciate the reviewer pointing out the confusion regarding the reference numbering in our previous response. We apologize for the lack of clarity. To comply with the journal's formatting guidelines, the Supplementary Information (SI) maintains a dedicated, independent reference list that starts from [1]. Consequently, the numbering in the SI does not align with the numbering in the Main Manuscript. In our previous response letter, citations [8] and [14] referred to the Main Manuscript's bibliography, whereas citations [10] and [11] referred to the SI bibliography. We understand that using these numbers interchangeably without specifying the source was confusing. We have ensured that the references are organized following the guidelines.

Reviewer #2 (Remarks to the Author):

The author provided detailed responses to the previous suggestions. After review, I believe that the authors have adequately addressed the issues raised and my previous concerns have been resolved. I suggest accepting the current form of the manuscript.

-> Response:

We sincerely thank the reviewer for the positive feedback and for recommending our manuscript for publication. We appreciate the time the reviewer dedicated to evaluating our work and the constructive suggestions provided during the review process.